# Extreme heterogeneity of influenza virus infection in single cells

**Alistair B Russell[1], Cole Trapnell[2], Jesse D Bloom[1,2]***

[1]Basic Sciences Division and Computational Biology Program, Fred Hutchinson Cancer Research Center, Seattle, United States; [2]Department of Genome Sciences, University of Washington, Seattle, United States

**Abstract** Viral infection can dramatically alter a cell's transcriptome. However, these changes have mostly been studied by bulk measurements on many cells. Here we use single-cell mRNA sequencing to examine the transcriptional consequences of influenza virus infection. We find extremely wide cell-to-cell variation in the productivity of viral transcription – viral transcripts comprise less than a percent of total mRNA in many infected cells, but a few cells derive over half their mRNA from virus. Some infected cells fail to express at least one viral gene, but this gene absence only partially explains variation in viral transcriptional load. Despite variation in viral load, the relative abundances of viral mRNAs are fairly consistent across infected cells. Activation of innate immune pathways is rare, but some cellular genes co-vary in abundance with the amount of viral mRNA. Overall, our results highlight the complexity of viral infection at the level of single cells.
DOI: https://doi.org/10.7554/eLife.32303.001

## Introduction

Viruses can cause massive and rapid changes in a cell's transcriptome as they churn out viral mRNAs and hijack cellular machinery. For instance, cells infected with influenza virus at high multiplicity of infection (MOI) express an *average* of 50,000 to 100,000 viral mRNAs per cell, corresponding to 5% to 25% of all cellular mRNA (*Hatada et al., 1989*). Infection can also trigger innate-immune sensors that induce the expression of cellular anti-viral genes (*Killip et al., 2015*; *Iwasaki and Pillai, 2014*; *Crotta et al., 2013*). This anti-viral response is another prominent transcriptional signature of high-MOI influenza virus infection in bulk cells (*Geiss et al., 2002*).

However, initiation of an actual influenza infection typically involves just a few virions infecting a few cells (*Varble et al., 2014*; *Poon et al., 2016*; *Sobel Leonard et al., 2017*; *McCrone et al., 2017*). The dynamics of viral infection in these individual cells may not mirror bulk measurements made on many cells infected at high MOI. Over 70 years ago, Max Delbruck showed that there was a ~100-fold range in the number of progeny virions produced per cell by clonal bacteria infected with clonal bacteriophage (*Delbrück, 1945*). Subsequent work has shown similar heterogeneity during infection with other viruses (*Zhu et al., 2009*; *Schulte and Andino, 2014*; *Combe et al., 2015*; *Akpinar et al., 2015*), including influenza virus (*Heldt et al., 2015*).

In the case of influenza virus infection, targeted measurements of specific proteins or RNAs have shed light on some factors that contribute to cell-to-cell heterogeneity. The influenza virus genome consists of eight negative-sense RNA segments, and many infected cells fail to express one more of these RNAs (*Heldt et al., 2015*; *Dou et al., 2017*) or their encoded proteins (*Brooke et al., 2013*). In addition, activation of innate-immune responses is inherently stochastic (*Shalek et al., 2013*; *Shalek et al., 2014*; *Bhushal et al., 2017*; *Hagai et al., 2017*), and only some influenza-infected cells express anti-viral interferon genes (*Pérez-Cidoncha et al., 2014*; *Killip et al., 2017*). However, the extent of cell-to-cell variation in these and other host and viral factors remains unclear, as does the association among them in individual infected cells.

**\*For correspondence:**
jbloom@fredhutch.org

**Competing interests:** The authors declare that no competing interests exist.

**eLife digest** When viruses infect cells, they take over the cell's machinery and use it to express their own genes. This process has mostly been studied by looking at the average outcome of infection when many viruses infect many cells. However, it is less clear what happens in individual cells. For example, does the virus take over every cell to make lots of viral gene products, or do some cells produce far more viral gene products than others?

Russell et al. have now used a new technique called single-cell RNA sequencing to look at how well influenza virus genes were expressed in hundreds of individual mammalian cells. The goal was to work out how the outcome of infection varied between different cells.

One way to quantify variability – also known as heterogeneity – is by using a statistical measure called the Gini coefficient. This statistic is often used to assess the inequality in incomes across a nation. In the hypothetical situation where everyone earned the same income, the Gini coefficient would equal zero; while if only one person had all the income and all others had none, the value would be very close to one. In reality, countries fall somewhere in between these two extremes. In the United States for instance, the Gini coefficient for income is 0.47. When Russell et al. worked out the Gini coefficient for the amount of viral genes expressed in different cells, the value was at least 0.64. This indicates that there is more unevenness in viral gene expression for influenza than there is income inequality in the United States.

So, what characterizes the "Bill Gates" cells and viruses that have the highest viral gene expression? Influenza viruses sometimes fail to express some of their genes. Russell et al. found that this failure often led to "poor" viruses that were less productive than "rich" viruses that expressed all the critical genes. However, the results suggest that there are also other factors that contribute a lot to the heterogeneity.

Real influenza virus infections are usually started by very few viruses, so this new understanding of the variability that occurs when individual viruses infect individual cells might prove important for understanding the properties of infections at larger scales too.

DOI: https://doi.org/10.7554/eLife.32303.002

Here we use single-cell mRNA sequencing to quantify the levels of all cellular and viral mRNAs in cells infected with influenza virus at low MOI. We find extremely large variation in the amount of viral mRNA expressed in individual cells. Both co-infection and activation of innate-immune pathways are rare in our low-MOI infections, and do not appear to be the major drivers of cell-to-cell heterogeneity in viral transcriptional load. Individual infected cells often fail to express specific viral genes, and such gene absence explains some but certainly not all of the cell-to-cell heterogeneity. A variety of cellular genes, including ones involved in the oxidative-stress response, co-vary with viral transcriptional load. Overall, our work demonstrates remarkable heterogeneity in the transcriptional outcome of influenza virus infection among nominally identical cells infected with a relatively pure population of virions.

## Results

### Strategy to measure mRNA in single virus-infected cells

We performed single-cell mRNA sequencing using a droplet-based system that physically isolates individual cells prior to reverse transcription (*Zheng et al., 2017*; *Macosko et al., 2015*; *Klein et al., 2015*). Each droplet contains primers with a unique *cell barcode* that tags all mRNAs from that droplet during reverse-transcription. Each primer also contains a *unique molecular identifier (UMI)* that is appended to each mRNA molecule during reverse transcription. The 3' ends of the mRNAs are sequenced and mapped to the human and influenza virus transcriptomes to determine transcript identities. This information is combined with that provided by the UMIs and cell barcodes to quantify the number of molecules of each mRNA species that have been captured for each cell.

Infected cells will express viral as well as cellular mRNAs – however the cell barcodes and UMIs cannot distinguish whether a cell was initially infected by one or multiple viral particles. We therefore engineered an influenza virus (strain A/WSN/1933) that additionally carried *viral barcodes* consisting

of synonymous mutations near the 3' end of each transcript (*Figure 1A*). Critically, these synonymous mutations did not greatly impact viral growth kinetics (*Figure 1B*). We infected A549 human lung carcinoma cells with an equal mix of the wild-type and synonymously barcoded viruses. Cells infected by a single virion will exclusively express mRNAs from either wild-type or synonymously barcoded virus, whereas cells that are co-infected with multiple virions will often express mRNAs from both the wild-type and synonymously barcoded viruses (*Figure 1C*).

We took care to generate stocks of virus that were relatively 'pure' of defective particles. Stocks of viruses typically contain an array of biologically active viral particles, some of which are defective for replication owing to mutations or deletions in essential viral genes (*von Magnus, 1954*; *Huang and Baltimore, 1970*; *Brooke, 2014*; *Fonville et al., 2015*; *Lauring and Andino, 2010*;

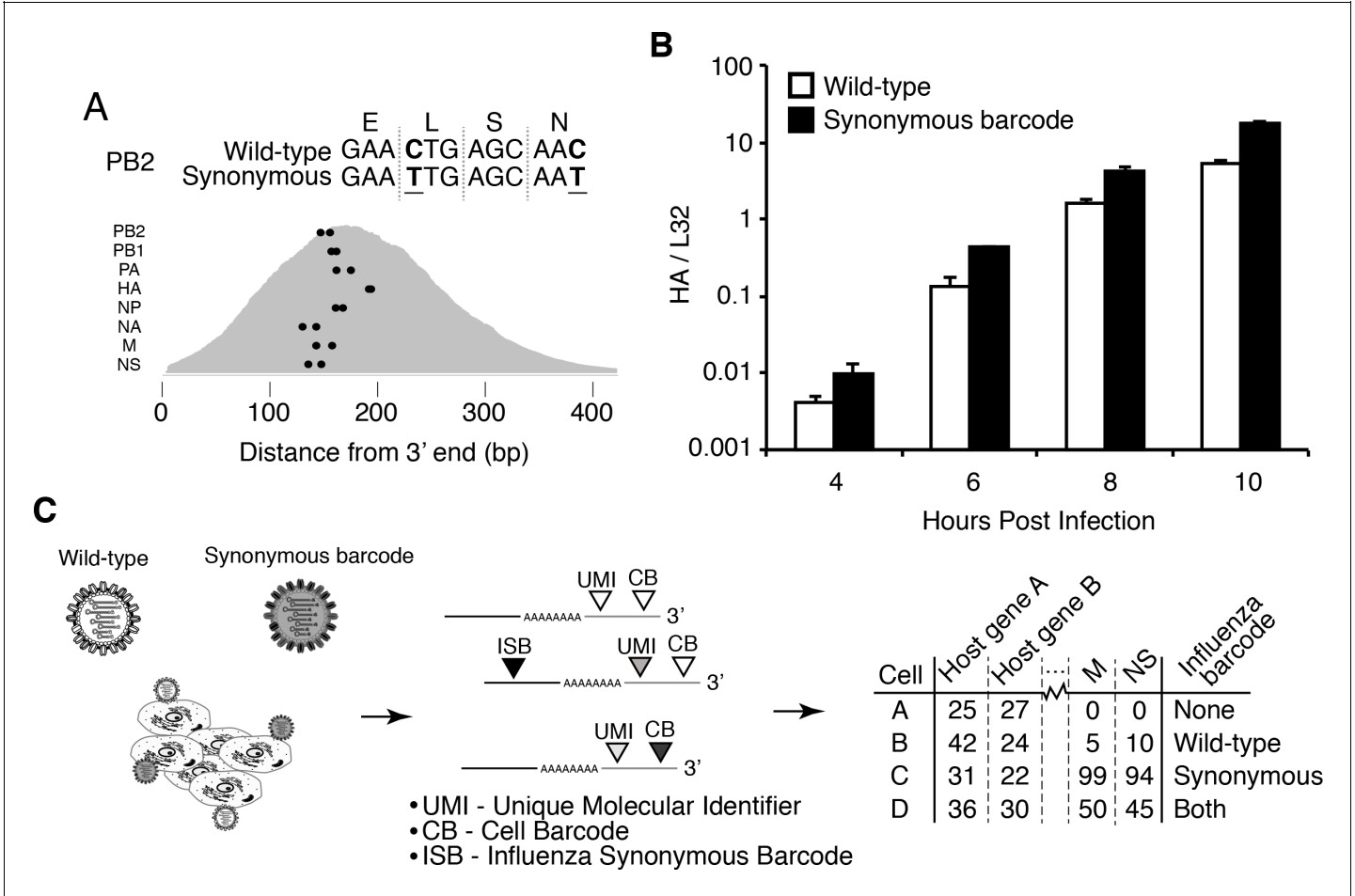

**Figure 1.** Experimental design. (**A**) We engineered a virus that carried two synonymous mutations near the 3' end of each mRNA. At top are the mutations for PB2. At bottom are locations of the synonymous mutations relative to the typical distribution of read depth for our 3'-end sequencing. (**B**) The wild-type and synonymously barcoded viruses transcribe their genes with similar kinetics. The abundance of the viral hemagglutinin (HA) transcript relative to the cellular housekeeping gene L32 was assessed by qPCR in A549 cells infected at an MOI of 0.5 (as determined on MDCK-SIAT1 cells). Error bars ± S.D., n = 3. (**C**) For the single-cell mRNA sequencing, A549 cells were infected with an equal mixture of wild-type and synonymously barcoded virus. Immediately prior to collection, cells were physically separated into droplets and cDNA libraries were generated containing the indicated barcodes. The libraries were deep sequenced, and the data processed to create a matrix that gives the number of molecules of each transcript observed in each cell. Infected cells were further annotated by whether their viral mRNAs derived from wild-type virus, synonymously barcoded virus, or both.

DOI: https://doi.org/10.7554/eLife.32303.003

The following source data is available for figure 1:

**Source data 1.** Sequences of wild-type and barcoded viruses are in viralsequences.fasta.
DOI: https://doi.org/10.7554/eLife.32303.004

*Dimmock et al., 2014*; *Saira et al., 2013*). These defective particles become prevalent when a virus is grown at high MOI, where complementation permits the growth of otherwise deleterious genotypes. To minimize the levels of defective particles, we propagated our viral stocks at low MOI for a relatively brief period of time (*Xue et al., 2016*). We validated that our stocks exhibited greater purity of infectious particles than a stock propagated at high MOI by verifying that they had a higher ratio of infectious particles to virion RNA (*Figure 2A*) and to particles capable of inducing expression of a single viral protein (*Figure 2B*). In addition, viral stocks with many defective particles are more immunostimulatory per infectious unit (e.g., TCID50) than low-defective stocks (*Tapia et al., 2013*; *López, 2014*), in part simply because there are more physical virions per infectious unit (*Figure 2A, B*). We confirmed that our viral stocks induced less interferon per infectious unit than a stock propagated at higher MOI (*Figure 2C*).

## Single cells show an extremely wide range of expression of viral mRNA

We infected A549 cells at low MOI with a mixture of the wild-type and synonymously barcoded viruses, and collected cells for sequencing at 6, 8, and 10 hr post-infection, performing two slightly different variants of the experiment for the 8 hr timepoint. For most of the samples, we replaced the infection inoculum with fresh media at one-hour post-infection, thereby ensuring that most infection was initiated during a narrow time window. However, for the second 8 hr sample (which we denote as '8 hr-2' in the figures), we did *not* perform this media change and instead left the cells in the original infection inoculum. The rationale for including a sample without a media change was to determine the importance of synchronicity of the timing of infection as discussed later in this subsection.

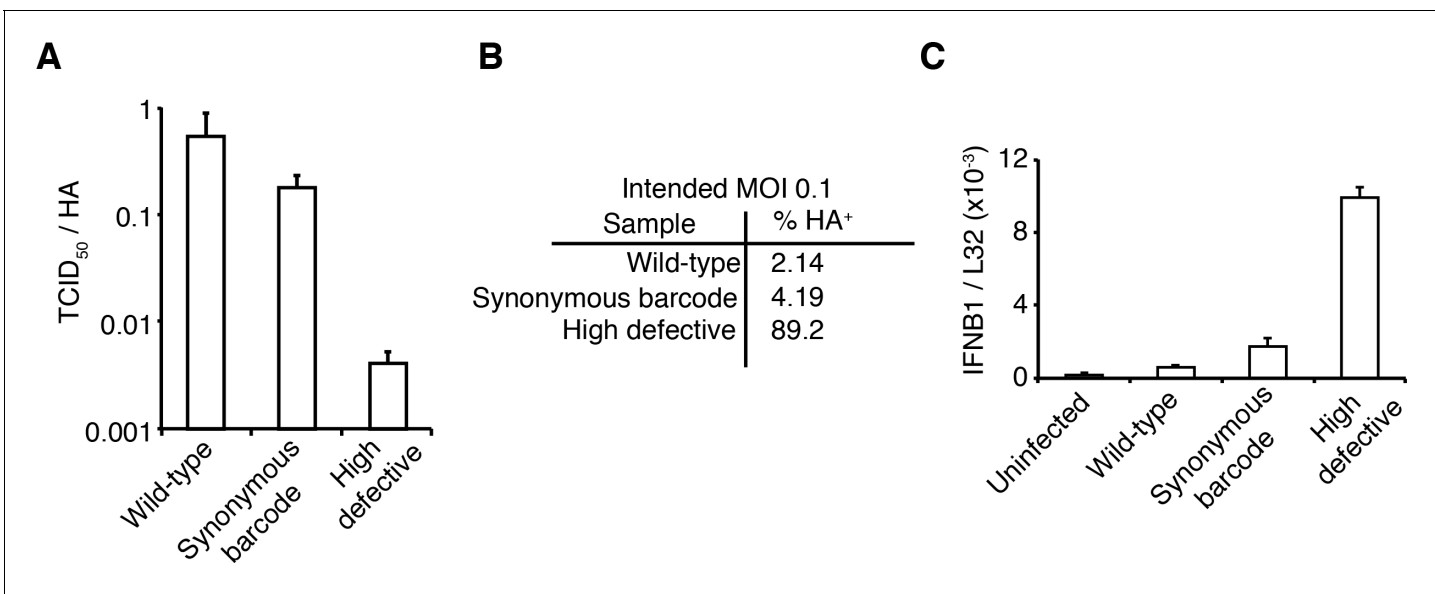

**Figure 2.** The viral stocks in our experiments are relatively pure of defective particles. (**A**) Our viral stocks have a higher ratio of infectious particles to HA virion RNA compared to a high-defective stock propagated at high MOI. HA viral RNA was quantified by qPCR on virions. Error bars ± S.D., n = 6 (qPCR replicates). (**B**) Our viral stocks have a higher ratio of infectious particles to particles capable of expressing HA protein. A549 cells were infected at an MOI of 0.1, and the percentage of cells expressing HA protein at 9 hr post-infection was quantified by antibody staining and flow cytometry. (**C**) Our viral stocks are less immunostimulatory than virus propagated at high MOI when used at the same number of infectious units as calculated by TCID50. Note that this fact does not necessarily imply that they are more immunostimulatory per virion, as the high-MOI stocks also have more virions per infectious unit as shown in the first two panels. Measurements of *IFNB1* transcript by qPCR normalized to the housekeeping gene L32 in A549 cells at 10 hr post infection at an MOI of 0.5. Error bars ± S.D., n = 3. Note that MOIs were calculated by TCID50 on MDCK-SIAT1 cells, whereas the experiments in this figure involved infection of A549 cells.

DOI: https://doi.org/10.7554/eLife.32303.005

The following figure supplement is available for figure 2:

**Figure supplement 1.** Full flow cytometry data for *Figure 2B*.

DOI: https://doi.org/10.7554/eLife.32303.006

We recovered between 3000 and 4,000 cells for each sample (*Figure 3A*). As expected for a low-MOI infection, most cells expressed little or no viral mRNA (*Figure 3B*, *Figure 3—figure supplement 1*). Also as expected, the amount of viral mRNA per cell among infected cells increased over time (*Figure 3B*, *Figure 3—figure supplement 1*). But what was most notable was how widely the number of viral mRNA molecules varied among infected cells. While the fraction of mRNA derived from virus was <0.1% for most cells, viral mRNA constituted half the transcriptome in a few cells at 8 and 10 hr (*Figure 3B*, *Figure 3—figure supplement 1*).

A complicating factor is that uninfected cells could have small amounts of viral mRNA due to leakage of transcripts from lysed cells. It is therefore important to establish a threshold for identifying truly infected cells. We can do this by taking advantage of the fact that roughly half the infecting virions bear synonymous barcodes. Reads derived from lysed cells will be drawn from both wild-type and synonymously barcoded viral transcripts. However, most cells are infected by at most one virion, and so the reads from truly infected cells will usually derive almost entirely from one of the two viral variants. *Figure 4A* shows the fraction of viral reads in individual cells from each viral variant, and *Figure 4B* indicates the fraction of viral reads from the most abundant variant in that cell. Most cells with large amounts of viral mRNA have viral transcripts exclusively derived from one viral variant – indicating non-random partitioning as expected from viral infection. However, cells with a small amount of viral mRNA often have viral transcripts from both variants, as expected from the random partitioning associated with simple mRNA leakage. Finally, a few cells with large amounts of viral mRNA have viral transcripts from both variants, likely reflecting co-infection.

We determined the threshold amount of viral mRNA per cell for each sample at which the barcode partitioning clearly resulted from infection rather than leakage (*Figure 4C*, *Figure 4—figure supplement 2*), and used these thresholds to annotate cells that we were confident were truly infected. We also annotated as co-infected cells above this threshold that had mRNA from both viral variants. *Figure 4D* shows the number of cells annotated as infected and co-infected for each sample – these cells are just a small fraction of the number of cells with any viral read. These annotation thresholds are conservative, and may miss some true low-level infections. However, it is important that the analyses below are restricted to cells that are truly infected with virus, so we accepted the possible loss of some low-level infections in order to avoid false positives. In addition, the synonymous viral barcodes only identify co-infections by viruses with different barcodes – since the barcodes are at roughly equal proportion, we expect to miss about half of the co-infections. Since we annotate about ~10% of the infected cells as co-infected by viruses with different barcodes

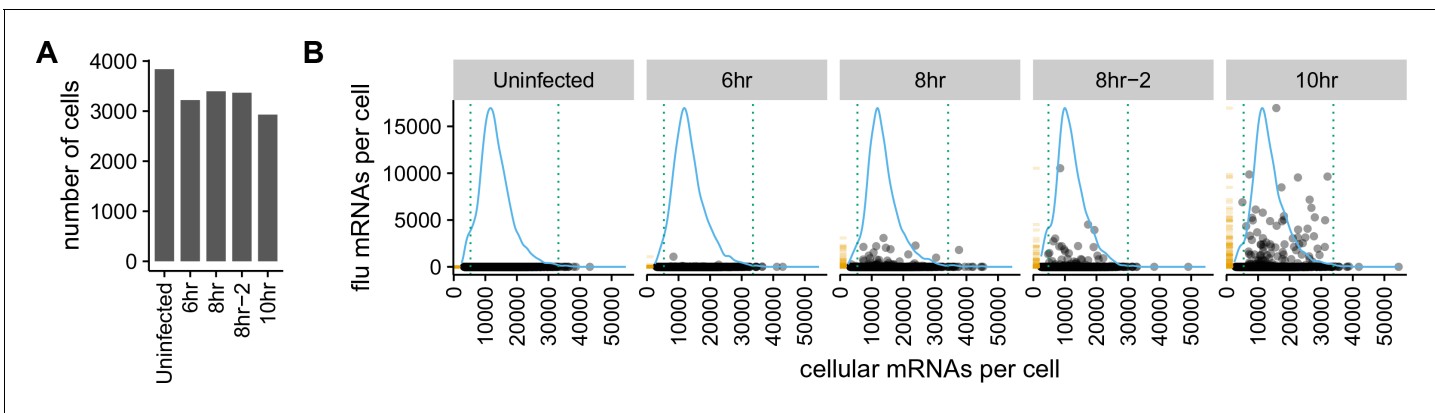

**Figure 3.** There is a very wide distribution in the amount of viral mRNA per cell. (**A**) Number of cells sequenced for each sample. (**B**) The number of cellular and viral mRNAs detected for each cell is plotted as a point. The blue lines show the overall distribution of the number of cellular mRNAs per cell. The orange rug plot at the left of each panel shows the distribution of the number of viral mRNAs per cell. Cells outside the dotted green lines were considered outliers with suspiciously low or high amounts of cellular mRNA (possibly derived from two cells per droplet), and were excluded from all subsequent analyses. *Figure 3—figure supplement 1* shows the exact distributions of the fraction of viral mRNA per cell.

DOI: https://doi.org/10.7554/eLife.32303.007

The following figure supplement is available for figure 3:

**Figure supplement 1.** Cumulative fraction plot of proportion of total mRNA from virus.

DOI: https://doi.org/10.7554/eLife.32303.008

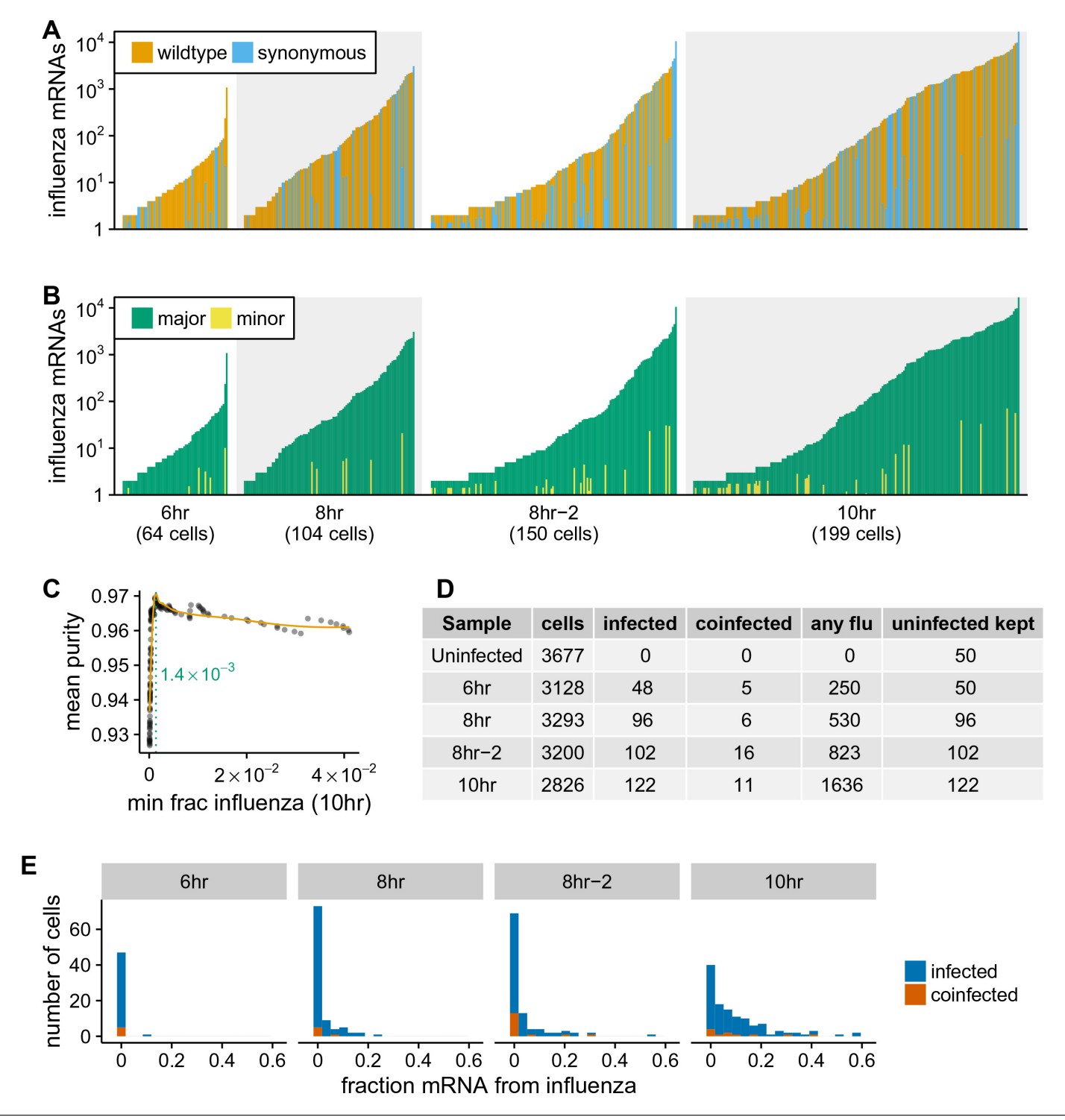

**Figure 4.** Synonymous barcodes on the viral mRNAs distinguish true infections from cells that contain viral mRNAs derived from leakage of lysed cells. (A) Cells with at least two viral mRNAs for which the barcode could be called, arranged in order of increasing influenza transcript counts. Bar heights denote the number viral mRNAs on a log10 scale, bar coloring is linearly proportional to the fractions of viral mRNAs derived from wild-type and synonymously barcoded virus. (B) Same as (A), but each bar is colored according to the relative fraction of the more common (major) and less common (minor) virus variant. At low levels of viral mRNA there is often a roughly equal mix, suggesting contamination with viral mRNAs leaked from lysed cells. At higher levels of viral mRNA, cells generally have only one viral variant, suggesting infection initiated by a single virion. A few cells are also obviously co-infected with both viral variants. (C) We determined a threshold for calling 'true' infections by finding the amount of viral mRNA per cell at which the viral barcode purity no longer increases with more viral mRNA. The purity is the fraction of all viral mRNA in a cell derived from the most abundant viral

*Figure 4 continued on next page*

*Figure 4 continued*

barcode in that cell. We fit a curve (orange line) to the mean purity of all cells with more than the indicated amount of viral mRNA, and drew the cutoff (dotted green line) at the point where this curve stopped increasing with the fraction of total mRNA derived from virus. This plot illustrates the process for the 10 hr sample, see *Figure 4—figure supplement 2* for similar plots for other samples. See the Materials and methods for details. (D) The number of cells identified as infected and co-infected for each sample, as well as the number of cells with any viral read. For all subsequent analyses, we subsampled the number of uninfected cells per sample to the greater of 50 or the number of infected cells. (E) Distribution of the fraction of mRNA per cell derived from virus for both infected and co-infected cells. *Figure 4—figure supplement 3* shows these same data in a cumulative fraction plots and calculates Gini coefficients to quantify the heterogeneity in viral mRNA load.

DOI: https://doi.org/10.7554/eLife.32303.009

The following figure supplements are available for figure 4:

**Figure supplement 1.** The number of viral barcodes called for each sample and gene segment.
DOI: https://doi.org/10.7554/eLife.32303.010
**Figure supplement 2.** Thresholds for calling infected cells.
DOI: https://doi.org/10.7554/eLife.32303.011
**Figure supplement 3.** Cumulative distributions of viral mRNA per cell and Gini coefficients.
DOI: https://doi.org/10.7554/eLife.32303.012
**Figure supplement 4.** Synchronization of infection does not greatly affect heterogeneity.
DOI: https://doi.org/10.7554/eLife.32303.013
**Figure supplement 5.** Effects of infectious dose or coinfection state.
DOI: https://doi.org/10.7554/eLife.32303.014

(*Figure 4D*), we expect another ~10% of the infected cells to also be co-infected but not annotated as so by our approach. Because most cells are not infected, we subsampled the uninfected cells to the numbers shown in *Figure 4D* to balance the proportions of infected and uninfected cells for all subsequent analyses.

Strikingly, the extreme variation in the number of viral transcripts per cell remains even after we apply these rigorous criteria for annotating infected cells (*Figure 4E*). The fraction of viral mRNA per infected cell follows a roughly exponential distribution, with many cells having few viral transcripts and a few cells having many. At 6 and 8 hr <10% of infected cells are responsible for over half the viral transcripts, while at 10 hr <15% of infected cells produce over half the viral transcripts (*Figure 4—figure supplement 3*). One way to quantify the heterogeneity of a distribution is to calculate the Gini coefficient (*Gini, 1921*), which ranges from 0 for a completely uniform distribution, to one for a maximally skewed distribution. *Figure 4—figure supplement 3* shows the Gini coefficients for the distribution of viral mRNA across infected cells for each sample. The Gini coefficients are ≥0.64 for all samples. As a fun point of comparison, these Gini coefficients indicate that the distribution of viral mRNA across infected cells is more uneven than the distribution of income in the United States (*Alvaredo, 2011*).

One possible source of heterogeneity in the amount of viral mRNA per cell is variability in the timing of infection. If some cells are infected earlier in the experiment than others, then they might have substantially more viral mRNA. However, several lines of evidence indicate that this is not the major cause of heterogeneity across cells. First, the sample for which the infection inoculum was never removed (8 hr-2) only shows slightly more heterogeneity than samples for which the inoculum was washed away after one hour (*Figure 4E*, *Figure 4—figure supplement 3*), despite the fact that the potential time window for infection is much longer in the former sample. Second, in an independent experiment, we performed completely synchronized infections by pre-binding virus to cells on ice and then washing away unbound virus before bringing the cells to 37°C (*Dapat et al., 2014*). As shown in *Figure 4—figure supplement 4*, flow cytometry staining found that the heterogeneity in the levels of individual viral proteins was not markedly different for these synchronized infections than in the absence of pre-binding and washing. Finally, viral mRNA expression from the secondary spread of virus from infected cells does not appreciably occur during the timeframes of our experiments, since *Figure 4B* does not show the pervasive presence of mixed barcodes that would occur in this case. Therefore, variability in the timing of infection is not the dominant cause of the cell-to-cell heterogeneity in our experiments.

Notably, *Figure 4E* shows that there are co-infected cells with both low and high amounts of viral mRNA, suggesting that the initial infectious dose does not drive a simple continuous increase in viral

transcript production. In support of this view, we used flow cytometry to quantify the levels of individual viral proteins in cells infected at various MOIs or for which we could delineate co-infection status (*Figure 4—figure supplement 5*). This analysis shows that sub-populations of cells that express similarly low and high levels of viral proteins persist across a wide range of infectious doses, although co-infection can influence the relative proportion of infected cells that fall into these sub-populations (*Figure 4—figure supplement 5*).

## Absence of viral genes partially explains cell-to-cell variability in viral load

The influenza genome is segmented, and cells can fail to express a viral mRNA if the encoding gene segment is not packaged in the infecting virion or fails to initiate transcription after infection. Indeed, several groups have reported that the majority of infected cells fail to express at least one viral gene (*Brooke et al., 2013*; *Heldt et al., 2015*; *Dou et al., 2017*). We wondered if the absence of specific viral genes might be associated with reduced amounts of viral mRNA within single infected cells. In particular, transcription of influenza virus mRNAs is performed by the viral ribonucleoprotein (RNP) complex, which consists of the three proteins that encode the tripartite polymerase (PB2, PB1, and PA) as well as nucleoprotein (NP) (*Huang et al., 1990*). Each viral gene segment is associated with one RNP in incoming infecting virions, but secondary transcription by newly synthesized RNPs requires the presence of the viral genes encoding each of the four RNP proteins (*Vreede et al., 2004*; *Eisfeld et al., 2015*). This secondary transcription is a major source of viral mRNAs, as evidenced by the fact that blocking synthesis of the RNP proteins reduces the amount of viral mRNA by several orders of magnitude in bulk cells (*Figure 5—figure supplement 1*).

We examined the total amount of viral mRNA versus the expression of the genes from each viral segment (*Figure 5A*, *Figure 5—figure supplement 2*, *Figure 5—figure supplement 3*). Note that influenza virus expresses ten major gene transcripts from its eight gene segments, as the M and NS segments are alternatively spliced to produce the M1/M2 and NS1/NEP transcript, respectively (*Dubois et al., 2014*). However, an inherent limitation of current established single-cell mRNA sequencing techniques is that they only sequence the 3' end of the transcript (*Zheng et al., 2017*; *Macosko et al., 2015*; *Klein et al., 2015*; *Cao et al., 2017*). Since the alternative spliceoforms M1/M2 and NS1/NEP share the same 3' ends, we cannot distinguish them and therefore will refer simply to the combined counts of transcripts from each of these alternatively spliced segments as the M and NS genes.

Cells that lack an RNP gene never derive more than a few percent of their mRNAs from virus, confirming the expected result that all four RNP genes are essential for high levels of viral transcription (*Figure 5A*, *Figure 5—figure supplement 2*, *Figure 5—figure supplement 3*). However, we observe cells that lack each of the other non-RNP genes but still derive ≈40% of their mRNAs from virus, suggesting that none of the other genes are important for high levels of viral transcription. These results are statistically supported by *Figure 5B*, which shows that absence of any RNP gene but *not* any other viral gene is associated with reduced amounts of viral mRNA. However, gene absence clearly does not explain all of the variability in viral gene expression, since even cells expressing all viral genes exhibit a very wide distribution in the amount of viral mRNA that they express. Specifically, at both 8 and 10 hr, the amount of viral mRNA in individual cells expressing all eight viral genes still ranges from <1% to >50% (*Figure 5A*, *Figure 5—figure supplement 2*, *Figure 5—figure supplement 3*). Furthermore, the actual distribution of viral mRNA per infected cell (*Figure 4E*) does not match the mostly bi-modal shape expected under a simple model where RNP gene absence and Poisson co-infection are the only factors (*Figure 5—source data 2*), indicating that there are additional sources of variability beyond whether cells have full complement of RNP genes.

We also quantified the fraction of infected cells that completely failed to express a given gene. We limited this analysis to examining the presence/absence of the non-RNP genes in cells expressing all four RNP genes, since we might fail to detect viral transcripts that are actually present at low levels in RNP-deficient cells due to the lower viral burden in these cells. At the 8- and 10 hr time points, between 5% and 17% of cells fail to express any one of the four non-RNP genes (*Figure 5C*, *Figure 5—source data 1*). The absence of a given gene appears to be an independent event, as the probability of observing all four non-RNP genes in a cell is well predicted by simply multiplying the probabilities of observing each gene individually (*Figure 5C* and *Figure 5—source data 1*). If we

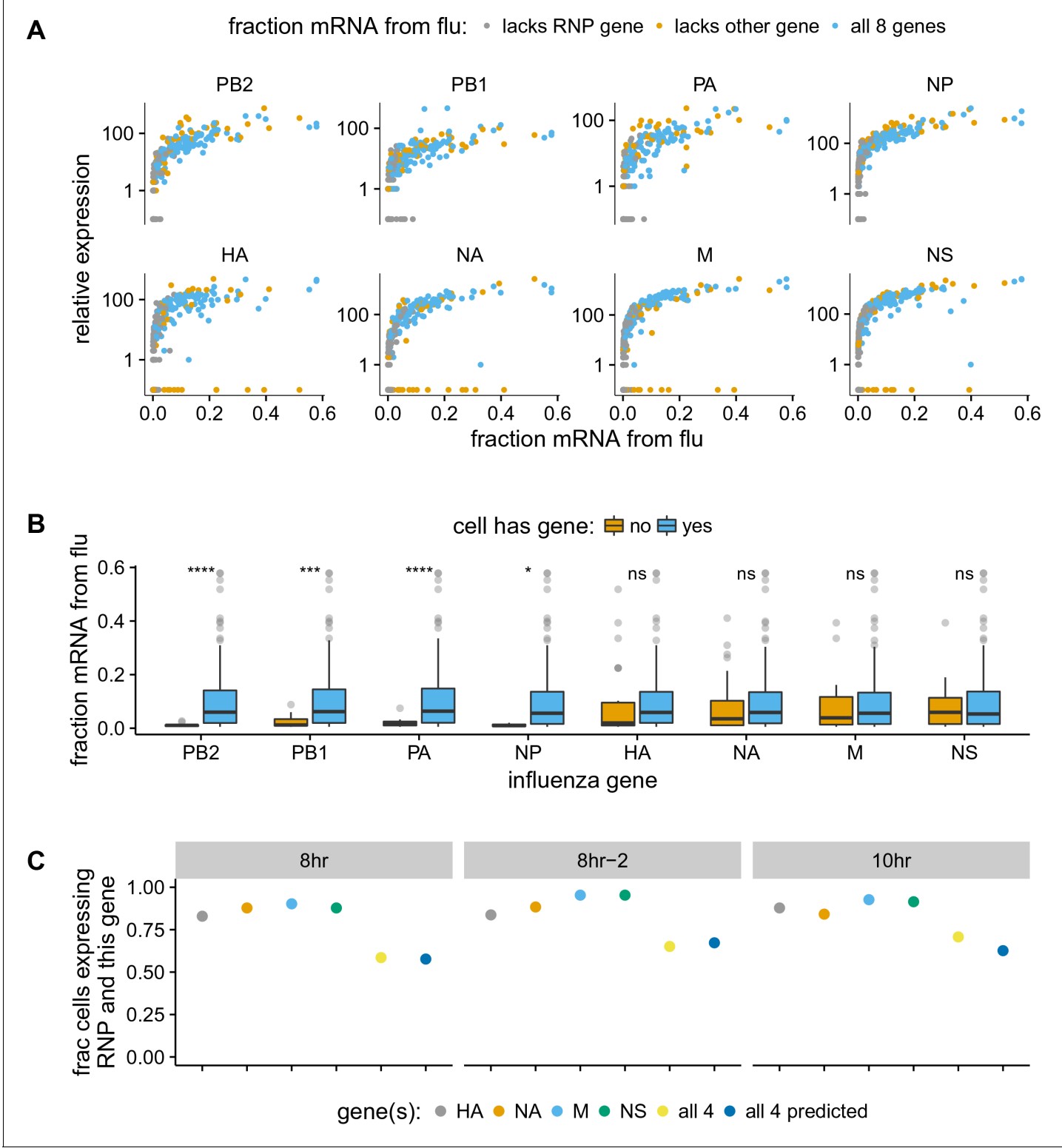

**Figure 5.** The absence of viral genes explains some of the variability in the amount of viral mRNA per cell. (**A**) The normalized expression of each viral gene as a function of the total fraction of mRNA in each infected cell derived from virus, taken over all time points. Cells with high viral burden always express all RNP genes, but some cells with high viral burden lack each of the other genes. Plots for individual samples are in *Figure 5—figure supplement 2*, and a plot that excludes known coinfected cells is in *Figure 5—figure supplement 3* . (**B**) Box and whisker plots showing the per-cell viral burden among cells with >0.5% of their mRNA from virus, binned by whether or not the cells express each gene. A Wilcoxon signed-rank test was used to test the null hypothesis that absence of each gene does not affect viral burden: **** = $P<10^{-4}$, *** = $P<10^{-3}$, * = $P<0.05$, ns = not significant. *Figure 5 continued on next page*

*Figure 5 continued*

The trends are similar if we look only at the 10 hr sample *Figure 5—figure supplement 4* or exclude known co-infected cells ([fluburdenbyflugene] fluburdenbyflugene_nocoinfection).*Figure 5—figure supplement 5* . (C) The fraction of cells that express each of the four other genes among cells that express all RNP genes, as well as the fraction that express *all* four of the other genes. The fraction that express all four genes is well predicted by simply multiplying the frequencies of cells that express each gene individually, indicating that gene absence is approximately independent across these genes.

DOI: https://doi.org/10.7554/eLife.32303.015

The following source data and figure supplements are available for figure 5:

**Source data 1.** The numerical data for panel (C) are in p_missing_genes.csv.
DOI: https://doi.org/10.7554/eLife.32303.021
**Source data 2.** Simulation with a simple model for the expected heterogeneity due to Poisson co-infection and presence/absence of the full RNP is in simple_Poisson_model.html.
DOI: https://doi.org/10.7554/eLife.32303.022
**Figure supplement 1.** Secondary transcription is a major source of viral mRNA during bulk infections.
DOI: https://doi.org/10.7554/eLife.32303.016
**Figure supplement 2.** Like panel (A), but shows samples individually.
DOI: https://doi.org/10.7554/eLife.32303.017
**Figure supplement 3.** Like panel (A), but excludes coinfected cells with mixed viral barcodes.
DOI: https://doi.org/10.7554/eLife.32303.018
**Figure supplement 4.** Like panel (B) but for the 10 hr sample only.
DOI: https://doi.org/10.7554/eLife.32303.019
**Figure supplement 5.** Like panel (B) but excludes coinfected cells with mixed viral barcodes.
DOI: https://doi.org/10.7554/eLife.32303.020

extrapolate the frequencies at which cells lack non-RNP genes to the RNP genes, then we would predict that 35–50% of infected cells express mRNAs from all eight genes. This estimate of the frequency at which infected cells express mRNAs from all eight gene segments is slightly higher than previous estimates of 13% (*Brooke et al., 2013*) and 20% (*Dou et al., 2017*). At least one difference is that Brooke et al. (*Brooke et al., 2013*) stained for proteins whereas we examined the expression of mRNAs – it is likely that some cells contain mutated viral genes that fail to produce stable protein even when mRNA is expressed.

## The relative amounts of different viral mRNAs are more consistent across cells

The results above show that the amount of viral mRNA in infected cells varies over several orders of magnitude. Does the relative expression of viral genes exhibit similar cell-to-cell variability? To address this question, we focused on cells that derived >5% of their mRNA from virus, since estimates of relative viral gene expression will be less noisy in cells with more viral mRNAs.

In contrast to the extreme variability in the total viral mRNA per cell, the fraction of this viral mRNA derived from each gene is much more consistent across cells (*Figure 6A*). Total viral mRNA varies by orders of magnitude, but the fraction from any given viral gene is fairly tightly clustered around the median value for all cells (*Figure 6B*). The relative levels of each viral mRNA in our cells are similar to prior bulk measurements made by Northern blots (*Hatada et al., 1989*), which also found an expression hierarchy of M > NS ≫ NP > NA > HA ≫ PB2 ~ PB1 ~ PA. The cell-to-cell consistency in the relative expression of different viral genes is even tighter if we limit the analysis only to cells that express all eight viral genes (*Figure 6C*). Therefore, with the exception of complete gene absence, the factors that drive the dramatic cell-to-cell variability in the amount of viral mRNA have roughly similar effects on all viral genes in a given cell. This finding is consistent with prior work showing positive correlations among the abundance of several viral genome segments in individual cells (*Heldt et al., 2015*).

## Co-infection can provide infected cells with the full complement of viral genes

Our sequencing enables us to identify the rare cells that were co-infected with both wild-type and synonymously barcoded viral variants. Overall, we captured 10 such co-infected cells that had >5%

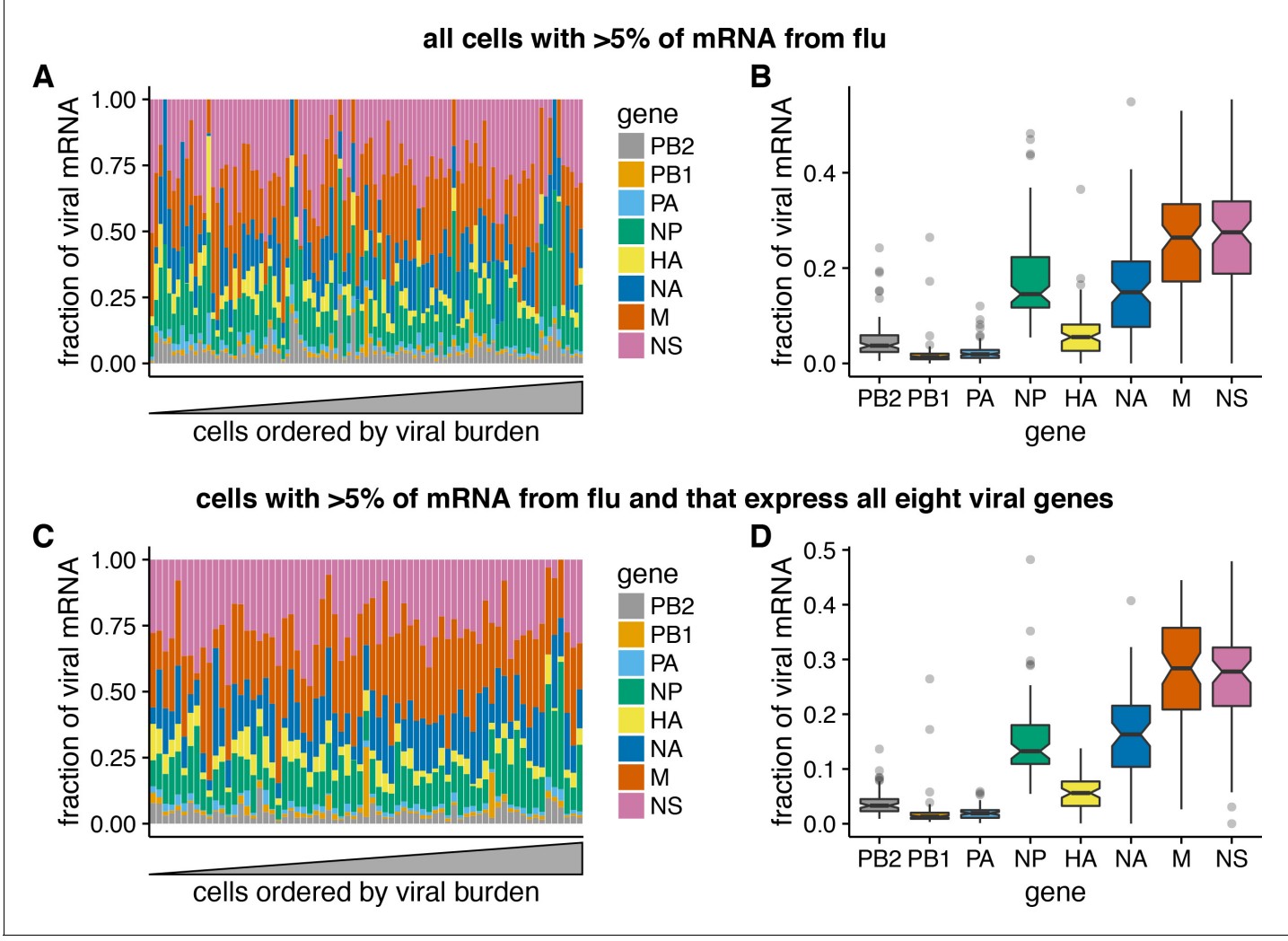

**Figure 6.** Relative expression of influenza virus genes in highly infected cells (>5% of total mRNA from virus). (**A**) The fraction of viral mRNA from each viral gene for each cell. (**B**) Box plots showing the distribution of the fraction of viral mRNA per cell from each viral gene. The black lines at the notches are the medians, and the tops and bottoms of boxes indicate the first and third quartiles. Whiskers extend to the highest or lowest data point observed within 1.5x the interquartile range, outliers shown as circles. Notches extend 1.58x the interquartile range divided by the square root of the number of observations. (**C**), (**D**) The same plots, but only including cells for which we observed at least one molecule of each viral gene.

DOI: https://doi.org/10.7554/eLife.32303.023

The following source data is available for figure 6:

**Source data 1.** The raw data for all cells are in p_flu_expr_all.csv.

DOI: https://doi.org/10.7554/eLife.32303.024

**Source data 2.** The raw data for fully infected cells are in p_flu_expr_fullyinfected.csv.

DOI: https://doi.org/10.7554/eLife.32303.025

of their mRNA derived from virus (*Figure 7*). Seven of these 10 cells expressed all eight viral genes. The majority (4 of 7) of these cells would *not* have expressed all the viral genes in the absence of co-infection, since they have at least one gene exclusively derived from each viral variant. For instance, the cell with 11.2% of its mRNA from virus in the upper right of *Figure 7* expresses M only from the wildtype viral variant, and NP and HA only from the synonymously barcoded variant. Our data therefore provide the first direct single-cell observation of the fact that co-infection can rescue missing viral genes (*Brooke et al., 2013*; *Brooke et al., 2014*; *Fonville et al., 2015*; *Aguilera et al., 2017*).

Another observation from *Figure 7* is that co-infected cells usually express roughly equal amounts of transcripts from each of the two viral variants. This observation is consistent with the finding by

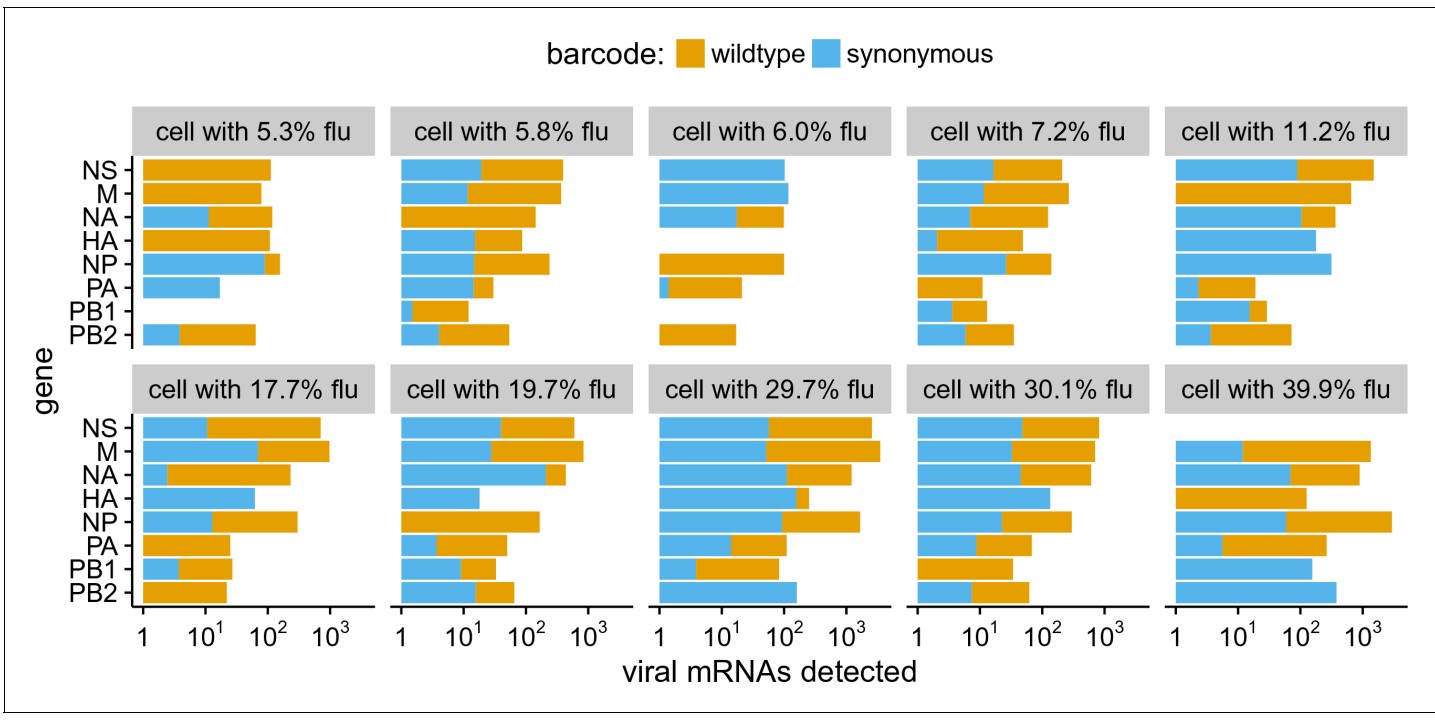

**Figure 7.** The abundance of each viral transcript in cells that are co-infected with the two viral variants and have >5% of their mRNA derived from virus. The bars show the logarithms of the numbers of each viral mRNA detected, and are colored in linear proportion to the fraction of that mRNAs derived from wild-type or synonymously barcoded virus.
DOI: https://doi.org/10.7554/eLife.32303.026

The following source data and figure supplement are available for figure 7:

**Source data 1.** The raw data plotted in this figure are in p_co-infection.csv.
DOI: https://doi.org/10.7554/eLife.32303.028

**Source data 2.** The sequence of the HA viral RNA carrying the GFP gene is in HAflank-eGFP.fasta.
DOI: https://doi.org/10.7554/eLife.32303.029

**Figure supplement 1.** Co-infected cells express roughly equal amounts of a gene from each infecting viral variant.
DOI: https://doi.org/10.7554/eLife.32303.027

*Doud et al., 2017* and *Huang et al. (2008)* that the temporal window for co-infection is short – if both viral variants infect a cell at about the same time, then neither will have a headstart and so each will have a roughly equal opportunity to transcribe its genes.

To support this idea with a larger dataset albeit at lower resolution, we generated a virus in which the HA coding sequence was replaced by GFP. We then co-infected cells with a mix of wildtype and ΔHA-GFP virus and used flow cytometry to score cells for the presence of HA only (infection by wild-type virus), GFP only (infection by ΔHA-GFP virus), or both (co-infection) as shown in *Figure 7—figure supplement 1*. As in our single-cell sequencing data, we found that expression of HA and GFP were highly correlated, indicating that co-infected cells typically expressed roughly equal amounts of transcript from each viral variant.

## Activation of the interferon response is rare in single infected cells

Because our sequencing captured all polyadenylated transcripts, we can examine whether there are prominent changes in the host-cell transcriptome in sub-populations of infected cells. Influenza virus infection can trigger innate-immune sensors that lead to the transcriptional induction of type I and III interferons, and subsequently of anti-viral interferon-stimulated genes (*Killip et al., 2015*; *Iwasaki and Pillai, 2014*; *Crotta et al., 2013*). However, activation of the interferon response is stochastic and bi-modal at the level of single cells (*Chen et al., 2010*; *Shalek et al., 2013*, *Shalek et al., 2014*; *Pérez-Cidoncha et al., 2014*; *Bhushal et al., 2017*; *Hagai et al., 2017*). We

therefore hypothesized that we might see two sub-populations of infected cells: one in which the interferon response inhibited viral transcription, and another in which the virus was able to express high levels of its mRNA by evading or blocking this response.

To examine whether there were distinct sub-populations of virus-infected cells, we used a semi-supervised t-SNE approach (*Van der Maaten and Hinton, 2008*) to cluster cells by genes that co-varied with viral infection status. As shown in *Figure 8A,B*, this approach effectively grouped cells by the amount of viral mRNA that they expressed. Sample-to-sample variation was regressed away during the clustering, as cells did not obviously group by time-point, with expected exception that the uninfected and 6 hr samples had few cells in the region of the plot corresponding to large amounts of viral mRNA (*Figure 8C*).

But to our surprise, we did not see a prominent clustering of infected cells into sub-populations as expected if the interferon response was strongly activated in some cells. To investigate further, we annotated each cell by the total number of type I and III interferon transcripts detected. Remarkably, only a single cell expressed detectable interferon (*Figure 8D*). We also examined interferon-stimulated genes, which are induced by autocrine and paracrine interferon signaling. *Figure 8E* shows expression of one such gene, IFIT1 (31). As with interferon itself, expression of IFIT1 was rare and most prominent in the single interferon-positive cell, presumably due to the higher efficiency of autocrine versus paracrine signaling. Notably, interferon and interferon-stimulated genes were also relatively ineffective at blocking viral transcription in the single cell in which they were potently induced, since >10% of the mRNA in this cell was derived from virus (*Figure 8A,B,D,E*).

We posited that the paucity of interferon induction might be due to the activity of influenza virus's major interferon antagonist, the NS1 protein (*García-Sastre et al., 1998*; *Hale et al., 2008*). We therefore identified cells that expressed substantial amounts of viral mRNA but lacked the NS gene (*Figure 8F*). Consistent with the idea that NS1 is important for suppressing interferon, the one interferon-positive cell lacked detectable expression of the NS gene. But other cells that lacked NS expression still failed to induce a detectable interferon response, despite often having a substantial amount of their mRNA derived from virus (*Figure 8*). This result is in line with other work showing that NS1-deficient influenza virus does not deterministically induce interferon (*Killip et al., 2017*; *Kallfass et al., 2013*). Therefore, many individual infected cells fail to activate innate-immune responses even when the virus lacks its major interferon antagonist.

## Some host genes co-vary with viral gene expression

We examined whether any host genes were differentially expressed in cells with more viral mRNA. We restricted this analysis to infected cells with all eight viral genes in order to focus on cellular genes that were associated with viral mRNA burden independent of effects due to the presence or absence of particular viral transcripts. We identified 43 cellular genes that co-varied with viral mRNA expression at a false discovery rate of 0.1 (*Figure 9*, *Figure 9—source data 1*).

A gene-set analysis shows that many cellular genes that are associated with the amount of viral mRNA are involved in the response to reactive oxygen species or hypoxia (*Figure 9—source data 2*). Genes known or suspected to be regulated by the Nrf2 master regulator in response to oxidative stress are often expressed at higher levels in cells with more viral mRNA (*Figure 9*). These genes produce proteins that are involved in detoxification of reactive oxygen species or resultant products, the management of misfolded proteins, the electron transport chain, or a general stress response (*Figure 9—figure supplement 1*). We additionally see reduced expression of the nitric oxide synthase interacting protein (NOSIP). Transient oxidative stress is known to occur during viral infection, and may act in a proviral fashion via MAPK activation driving vRNP export (*Amatore et al., 2015*). The antioxidant response is thought to be largely antiviral, potentially through inhibition of MAPK activity (*Lin et al., 2016*; *Sgarbanti et al., 2014*). To directly test the effect of transient oxidative stress, we compared the fraction of cells that expressed detectable viral protein when infected either with or without pre-treatment to suppress oxidative stress. *Figure 9—figure supplement 2* shows that the cells pre-treated with an antioxidant exhibited less frequent detectable expression of viral protein. These results, in conjunction with the differential expression test in *Figure 9* and the prior work mentioned above, suggest that oxidative stress acts in a proviral fashion.

The gene-set analysis also found that the amount of viral mRNA was associated with the expression of genes involved in the G2-M cell-cycle checkpoint (*Figure 9—source data 2*). The cell-cycle associated genes CCND3, MKI67, UBE2S, and CENPF are all expressed at significantly lower levels

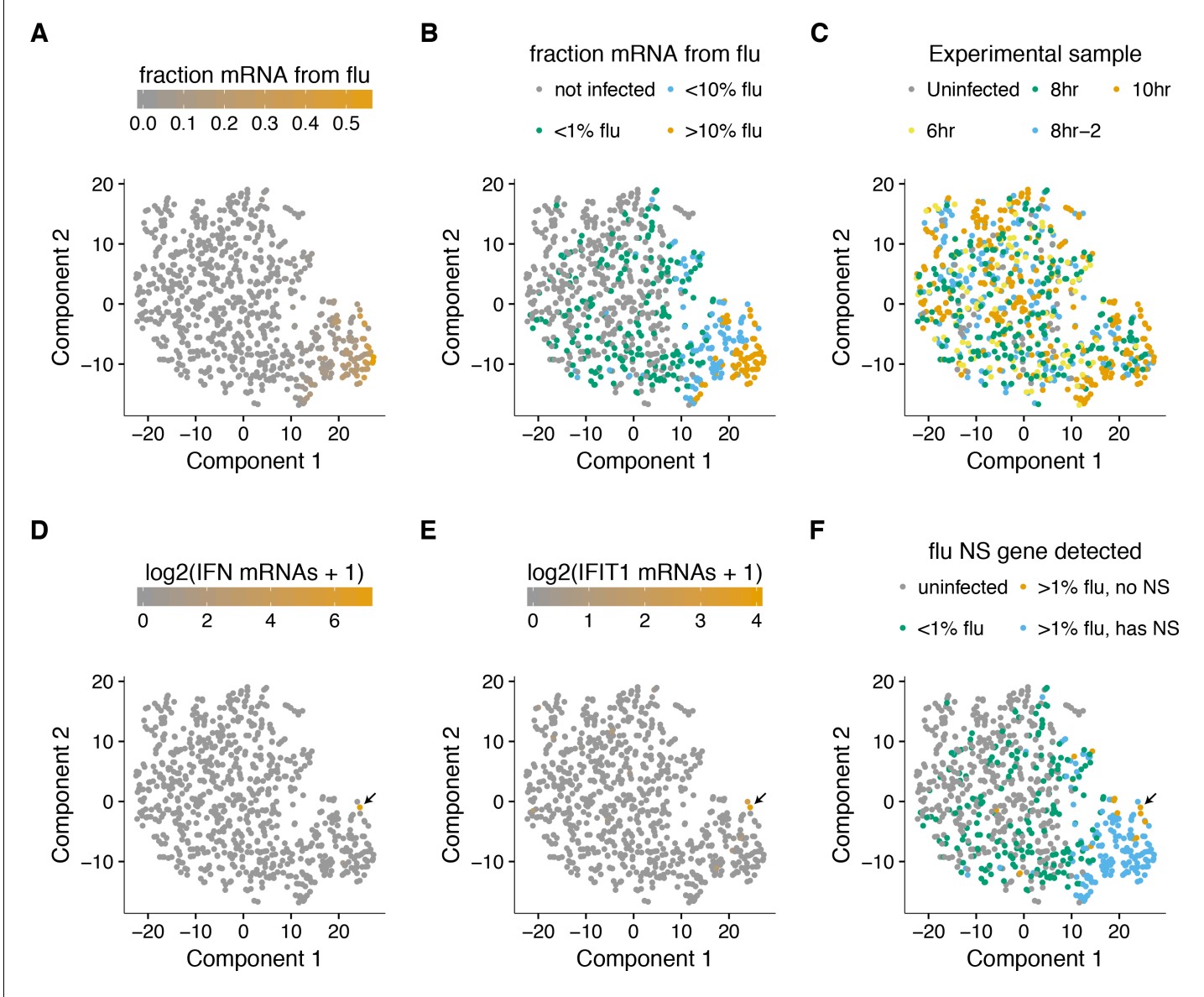

**Figure 8.** A t-SNE plot created by semi-supervised clustering using genes that co-vary with viral infection status. Each point is a single cell, and each panel shows an identical layout but colors the cells according to a different property. (A), (B) Cells colored by the fraction of their mRNA derived from virus. (C) Cells colored by the experimental sample. (D) Cells colored by the number of detected transcripts from type I and III interferons (IFN). Only one cell has detectable interferon expression (in orange, indicated with arrow). (E) Cells colored by the expression of the interferon-stimulated gene IFIT1. (F) Cells colored by whether they express the viral NS gene. The one interferon-positive cell is lacking NS, but so are many interferon-negative cells.

DOI: https://doi.org/10.7554/eLife.32303.030

in cells with more viral mRNA (*Figure 9*). However, our data are not sufficient to determine whether the lower expression of these genes is a cause or effect of the reduction in viral mRNA.

Interestingly, none of the cellular genes that are significantly associated with the amount of viral mRNA in our study are among the 128 genes that *Watanabe et al. (2010)* report as having been identified multiple times in genome-wide screens for factors affecting influenza virus replication. One possible explanation is that most of the cell-to-cell heterogeneity in our experiments might arise from viral segment absence or mutations, pure stochasticity, or more subtle alterations in host-cell state – not due to changes in expression of the type of single large-effect genes that are usually identified in genome-wide knockdown/knockout studies.

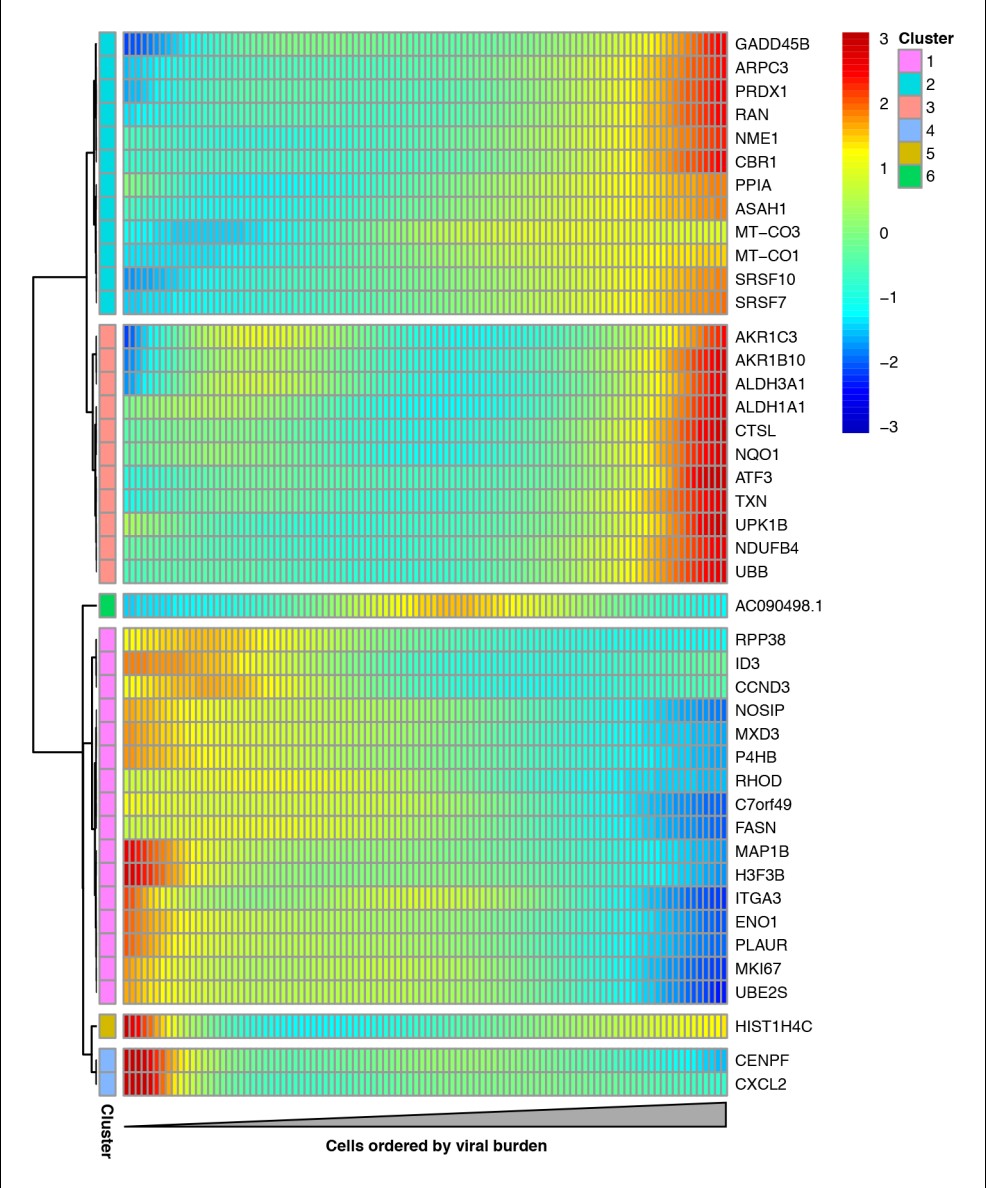

**Figure 9.** Cellular genes that co-vary in expression with the amount of viral mRNA in cells expressing all eight viral genes. The columns are cells, ordered from left to right by the fraction of mRNA derived from virus. Each row is a gene that is differentially expressed as a function of the fraction of mRNA derived from virus at a false discovery rate of 0.1. Genes for which the color goes from blue at left to red at right are expressed at higher levels in cells with more viral mRNA. The scale bar indicates the number of standard deviations above or below the mean expression, truncated at 3-fold on both sides.

DOI: https://doi.org/10.7554/eLife.32303.031

The following source data and figure supplements are available for figure 9:

**Source data 1.** The full results of the differential expression test are in p_sig_cellular_genes.csv.
DOI: https://doi.org/10.7554/eLife.32303.034

**Source data 2.** A gene-set analysis for pathways associated with the amount of viral mRNA is in p_pathway_enrichment.csv.
DOI: https://doi.org/10.7554/eLife.32303.035

**Figure supplement 1.** Many genes that co-vary with viral load are involved in the oxidative stress response.
DOI: https://doi.org/10.7554/eLife.32303.032

**Figure supplement 2.** Pre-treating to reduce oxidative stress decreases the fraction of infected cells expressing detectable viral protein.

*Figure 9 continued on next page*

*Figure 9 continued*

DOI: https://doi.org/10.7554/eLife.32303.033

## Discussion

We have quantified the total transcriptome composition of single cells infected with influenza virus. While we observe a general increase in the amount of viral mRNA over time as expected from bulk measurements (*Hatada et al., 1989*; *Shapiro et al., 1987*), there is wide variation in viral gene expression among individual infected cells.

The most obvious form of heterogeneity is the complete failure of some infected cells to express one or more viral genes, which we estimate occurs in about half the infected cells in our experiments. The absence of some viral genes in some infected cells has been noted previously (*Brooke et al., 2013*; *Heldt et al., 2015*; *Dou et al., 2017*), and our work provides a holistic view by quantifying the total viral transcriptional load as a function of the level of each mRNA. We find that cells lacking expression of any of the four genes that encode the viral RNP express much less total viral mRNA, consistent with prior bulk studies (*Vreede et al., 2004*; *Eisfeld et al., 2015*). Interestingly, the *reason* some cells fail to express some viral genes remains unclear. The prototypical influenza virion packages one copy of each of the eight gene segments (*Noda et al., 2006*; *Hutchinson et al., 2010*), but some virions surely package fewer (*Brooke et al., 2014*). However, it is also possible that much of the viral gene absence is due to stochastic loss of viral RNPs after infection but prior to the initiation of viral transcription in the nucleus.

The absence of viral genes only partially explains the cell-to-cell variation in amount of viral mRNA, which still varies from <1% to >50% among cells expressing all the viral genes. It is likely that other viral genetic factors explain some of this remaining heterogeneity. The 3'-end sequencing strategy used in our experiments detects the presence of a viral gene, but does not identify whether that gene contains a mutation that might hinder viral replication. However, viral mutations are also unlikely to explain all the observed heterogeneity, since current consensus estimates of influenza virus's mutation rate suggest that the typical virion in a stock such as the one used in our experiment should contain less than one mutation per genome (*Parvin et al., 1986*; *Suárez et al., 1992*; *Suárez-López and Ortín, 1994*; *Nobusawa and Sato, 2006*; *Bloom, 2014*; *Pauly et al., 2017*).

The rest of the heterogeneity must be due to some combination of cellular factors and inherent stochasticity. Some features of the cellular transcriptome co-vary with the amount of influenza mRNA. In particular, the viral load in individual cells is associated with the expression of genes involved in response to cellular stresses, including oxidative stress. It will be interesting to determine if these cellular transcriptional signatures are simply a consequence of the stress imposed by viral replication, or if their stronger activation in some cells is a causative factor that promotes viral transcription. However, it also would not be surprising if a substantial amount of the cell-to-cell heterogeneity cannot be ascribed to pre-existing features of either the viral genome or cellular state. Apparently stochastic heterogeneity is a common feature of many processes at a single-cell level (*Cai et al., 2006*; *Raj et al., 2006*; *Buganim et al., 2012*; *Shalek et al., 2013*; *Avraham et al., 2015*) – especially when those processes are initiated by very small numbers of initial molecules (*Elowitz et al., 2002*), as is the case for low-MOI viral infection.

Our data do suggest that the factors driving the heterogeneity in viral transcriptional load exert relatively concordant effects on all viral genes in a given cell. Specifically, despite the extreme heterogeneity in total viral mRNA per cell, the relative levels of the viral mRNAs are reasonably consistent across cells, and generally reflective of classical bulk measurements (*Hatada et al., 1989*). Therefore, despite the stochasticity inherent in initiating transcription and replication of each gene from a single copy carried by the incoming virion, as long as a gene is not completely lost then the virus possesses mechanisms to control its relative expression (*Shapiro et al., 1987*; *Hatada et al., 1989*; *Perez et al., 2010*; *Heldt et al., 2012*; *Chua et al., 2013*).

One factor that surprisingly does *not* appreciably contribute to the heterogeneity in our experiments is activation of innate-immune interferon pathways. Only one of the hundreds of virus-infected cells expresses any detectable interferon, despite the fact that a number of cells fail to express the influenza-virus interferon antagonist NS1. It is known that interferon activation is stochastic at the level of single cells in response to both synthetic ligands (*Shalek et al., 2013*, *Shalek et al., 2014*;

*Bhushal et al., 2017*; *Hagai et al., 2017*) and actual infection (*Rand et al., 2012*; *Pérez-Cidoncha et al., 2014*; *Avraham et al., 2015*; *Killip et al., 2017*). But interferon expression is a prominent transcriptional signature of high-MOI influenza virus infection of bulk cells, including in the epithelial cell line and at the time-points used in our experiments (*Geiss et al., 2002*; *Sutejo et al., 2012*). So it is notable how rarely single cells express interferon. Interferon expression would surely be more common at later times or with a viral stock passaged at higher MOI, since paracrine interferon signaling (*Crotta et al., 2013*) and accumulation of defective viral particles enhance innate-immune detection (*Tapia et al., 2013*; *López, 2014*). However, the early events of physiological influenza infection involve just a few virions (*Varble et al., 2014*; *McCrone et al., 2017*), and so it is interesting to speculate whether rare events such as interferon activation during the first few cycles of viral replication could contribute to heterogeneity in the eventual outcome of infection.

Overall, our work shows the power and importance of characterizing cellular infection at the level of single cells (*Avraham et al., 2015*). Viral infection can involve heterogeneity in the genetic composition of the incoming virion, the host-cell state, the bi-modality of innate-immune activation, and the inherent stochasticity of molecular processes initiated by a single copy of each viral gene. In our experiments with short-timeframe and low-MOI infections with a relatively pure stock of influenza virus, we find only a minor role for innate-immune activation, but a substantial role for heterogeneity in the complement of viral genes that are expressed in individual cells and at least some contribution of host-cell state. Our current experiments are not able to quantify the role of other possibly important factors such as mutations in viral genes, but we suspect that they may also contribute. Future extensions of the approaches described here should enable further deconstruction of the sources of cell-to-cell heterogeneity during viral infection, thereby enabling a deeper understanding of how the bulk features of infection emerge from processes within individual virus-infected cells.

## Materials and methods

### Cell lines and viruses

The following cell lines were used in this study: the human lung epithelial carcinoma line A549 (ATCC CCL-185), the MDCK-SIAT1 variant of the Madin Darby canine kidney cell line overexpressing human SIAT1 (Sigma-Aldrich 05071502), and the human embryonic kidney cell line 293T (ATCC CRL-3216). The A549 cells were tested as negative for mycoplasma contamination by the Fred Hutch Genomics Core, and authenticated using the ATCC STR profiling service. All cells were maintained in D10 media (DMEM supplemented with 10% heat-inactivated fetal bovine serum, 2 mM L-glutamine, 100 U of penicillin/ml, and 100 µg of streptomycin/ml) at 37 at 5% CO2.

Wildtype A/WSN/1933 (H1N1) influenza virus was generated by reverse genetics using the plasmids pHW181-PB2, pHW182-PB1, pHW183-PA, pHW184-HA, pHW185-NP, pHW186-NA, pHW187-M, and pHW188-NS (*Hoffmann et al., 2000*). The sequences of the viral RNAs encoded in these plasmids are in *Figure 1—source data 1*. Reverse-genetics plasmids encoding the synonymously barcoded WSN virus were created by using site-directed mutagenesis to introduce two synonymous mutations near the 3' end of the mRNA for each viral gene. The sequences of the synonymously barcoded viral RNAs are in *Figure 1—source data 1*.

To generate viruses from these plasmids, we transfected an equimolar mix of all eight plasmids into cocultures of 293T and MDCK-SIAT1 cells seeded at a ratio of 8:1. At 24 hr post-transfection, we changed media from D10 to influenza growth media (Opti-MEM supplemented with 0.01% heat-inactivated FBS, 0.3% BSA, 100 U of penicillin/ml, 100 µg of streptomycin/ml, and 100 µg of calcium chloride/ml). At 48 hr post-transfection we harvested the virus-containing supernatant, pelleted cellular material by centrifugation at 300 x g's for 4 min, and stored aliquots of the clarified viral supernatant at −80 . We then titered thawed aliquots of viral by TCID50 on MDCK-SIAT1 cells, computing titers via the formula of *Reed and Muench (1938)*. To generate our 'high-purity' stocks of viruses for the single-cell sequencing experiments, we then infected MDCK-SIAT1 cells at an MOI of 0.01, and let the virus grow for 36 hr prior to harvesting aliquots that were again clarified by low-speed centrifugation, aliquoted, stored at −80 , and titered by TCID50. The high-MOI passage (high-defective particle) stock used in *Figure 2* was generated by instead passaging in MDCK-SIAT1 cells twice at an MOI of 1 for 48 hr.

For the experiments in *Figure 7—figure supplement 1*, we created a virus that carried an HA gene segment in which GFP replaced most of the HA coding sequence, following a scheme first described by Marsh et al. (*Marsh et al., 2007*). Briefly, we created a plasmid encoding a viral RNA with GFP in place of the HA coding sequence in the context of the pHH21 (*Neumann et al., 1999*) reverse-genetics plasmid, removing potential start codons upstream of the GFP (see *Figure 7— source data 1* for the sequence of the viral RNA). We then generated GFP-carrying virus by reverse-genetics in cells constitutively expressing HA (*Doud and Bloom, 2016*). To obtain sufficient titers, this HA-eGFP virus was expanded for 44 rather than 36 hr after initiating infection at an MOI of 0.01.

## qPCR

For the qPCR in *Figure 2* and *Figure 5—figure supplement 1*, A549 cells were seeded at $3 \times 10^5$ cells per well in a 6-well tissue culture plate in D10 the day prior to infection. On the day of infection, a single well was trypsinized and the cells were counted in order to determine the appropriate amount of virus to use to achieve the intended MOI. Immediately before infection, D10 was replaced with influenza growth media. For cells incubated with cyclohexamide, the compound was added to a final concentration of 50 µg/ml at the time of infection – previously confirmed to be sufficient to halt viral protein production (*Killip et al., 2014*). RNA was purified using the QIAGEN RNeasy plus mini kit following manufacturer's instructions. cDNA was synthesized using an oligoDT primer and the SuperScript III first-strand synthesis supermix from ThermoFisher using the manufacturer's protocol. Transcript abundance was measured using SYBR green PCR master mix, using a combined anneal/ extension step of 60 for one minute with the following primers: *HA*: 5'-GGCCCAACCACACA TTCAAC-3', 5'-GCTCATCACTGCTAGACGGG-3', *IFNB1*: 5'-AAACTCATGAGCAGTCTGCA-3', 5'- AGGAGATCTTCAGTTTCGGAGG-3', *L32*: 5'-AGCTCCCAAAAATAGACGCAC-3', 5'-TTCATAGCAG TAGGCACAAAGG-3'. Biological triplicates were performed for all samples.

For the measurements of viral genomic HA content in *Figure 2*, vRNA was harvested from 80 µl of viral supernatant by the addition of 600 µl of RLT plus before proceeding with the standard QIA-GEN RNeasy Plus Mini kit protocol. The cDNA was generated using SuperScript III first-strand synthesis supermix using the manufacturer's protocol, and using the universal vRNA primers of Hoffmann et al. (*Hoffmann et al., 2001*) with the modifications described in Xue et al. (*Xue et al., 2017*). The qPCR was then performed as for mRNA measurements. A standard curve was generated from three independent dilutions of the HA-encoding reverse genetics plasmid. All vRNA values represent three independent RNA extractions with two replicate qPCR measurements.

## Flow cytometry titering and analyses

To determine viral titers in terms of HA-expressing units and for the flow cytometry, A549 cells were seeded in a 6-well plate and infected as described above for the qPCR analyses. Cells were harvested by trypsinization, resuspended in phosphate-buffered saline supplemented with 2% heat-inactivated FBS, and stained with 10 µg/ml of H17-L19, a mouse monoclonal antibody confirmed to bind to WSN HA in a prior study (*Doud et al., 2017*). After washing in PBS supplemented with 2% FBS, the cells were stained with a goat anti-mouse IgG antibody conjugated to APC. Cells were then washed, fixed in 1% formaldehyde, and washed further before a final resuspension and analysis. We then determined the fraction of cells that were HA positive and calculated the HA-expressing units. For NS1 staining, cells stained for HA as described above were permeabilized using BD Cytofix/ Cytoperm following manufacturer's instructions, stained with anti-NS1 (GTX125990, Genetex) at 4.4 µg/ml, washed, stained with a goat anti-rabbit IgG antibody conjugated to Alexa Fluor 405, washed, and analyzed. To analyze the effect of N-acetylcysteine, the compound was added to cells in D10 9 hr prior to media change and infection, and included in infection media. Stocks of N-acetylcysteine were reconstituted immediately prior to use, and the pH of growth media supplemented with this compound was adjusted using sodium hydroxide. After channels were compensated and cells gated to exclude multiplets and debris in FlowJo, data were extracted using the R package flowCore (*Le Meur et al., 2007*) and analyzed using a custom Python script. Guassian kernel density estimates were obtained using the scipy stats package method, guassian_kde, using automatic bandwidth determination (*van der Walt et al., 2017*). For gating on NS1 positive cells, the percentage of influenza-infected cells was determined by HA staining alone, and the top quantile of NS1-stained cells matching that percentage were taken as the NS1 positive population.

## Infections for single-cell mRNA sequencing

Single-cell sequencing libraries were generated using the 10x Chromium Single Cell 3' platform (*Zheng et al., 2017*) using the V1 reagents.

All time points except for the second 8 hr sample (8 hr-2) were prepared on the same day. For the infections, A549 cells were seeded in a 6-well plate, with two wells per time point. A single well of cells was trypsinized and counted prior to initiation of the experiment for the purposes of calculating MOI. Wild-type and synonymously barcoded virus were mixed to an estimated ratio of 1:1 based on prior, exploratory, single-cell analyses (data not shown). At the initiation of our experiment, the wells for all time points were changed from D10 to influenza growth media. Cells were then infected with 0.3 HA-expressing units of virus per cell (as determined by flow cytometry). The infections were performed in order of time point: first the 10 hr time point, then the 8 hr, and then the 6 hr time point. At one hour after infection, the media for each time point was changed to fresh influenza growth media. Note that the HA-expressing units were calculated without this additional washing step, and so likely represent an overestimate of our final infectious dose (consistent with the fact that fewer than 30% of cells appear infected in the single-cell sequencing data). All cells were then harvested for single-cell analysis concurrently – ensuring all had spent equivalent time in changed media . For 8 hr-2 sample, cells were infected as above except that the cells were infected at 0.1 HA-expressing units of virus per cell but no wash step was performed, and the sample was prepared on a different day. After harvest, cells were counted using disposable hemocytometers and diluted to equivalent concentrations with an intended capture of 3000 cells/sample following the manufacturer's provided by 10x Genomics for the Chromium Single Cell platform. All subsequent steps through library preparation followed the manufacturer's protocol. Samples were sequenced on an Illumina HiSeq.

## Computational analysis of single-cell mRNA sequencing data

Jupyter notebooks that perform all of the computational analyses are available in *Supplementary file 1* and at https://github.com/jbloomlab/flu_single_cell (*Russell et al., 2018*) copy archived at https://github.com/elifesciences-publications/flu_single_cell).

Briefly, the raw deep sequencing data were processed using the 10X Genomics software package CellRanger (version 2.0.0). The reads were aligned to a concatenation of the human and influenza virus transcriptomes. The human transcriptome was generated by filtering genome assembly GRCh38 for protein coding genes defined in the GTF file GRCh38.87. The influenza virus transcriptome was generated from the reverse-complement of the wildtype WSN viral RNA sequences as encoded in the reverse-genetics plasmids (*Figure 1—source data 1*). The aligned deep sequencing data are available on the GEO repository under accession GSE108041 (https://www.ncbi.nlm.nih.gov/geo/query/acc.cgi?acc=GSE108041).

CellRanger calls cells based on the number of observed cell barcodes, and creates a cell-gene matrix. We used custom Python code to annotate the cells in this matrix by the number of viral reads that could be assigned to the wildtype and synonymously barcoded virus. Only about half of the viral reads overlapped the barcoded regions of the genes (*Figure 1A*) and could therefore be assigned to a viral barcode (*Figure 4—figure supplement 1*). So for calculations of the number of reads in a cell derived from each viral barcode for each viral gene, the total number of detected molecules of that gene are multiplied by the fraction of those molecules with assignable barcodes that are assigned to that barcode. This annotated cell-gene matrix is in *Supplementary file 2*. A Jupyter notebook that performs these analyses is in *Supplementary file 1*.

The annotated cell-gene matrix was analyzed in R, primarily using the Monocle package (version 2.4.0) (*Qiu et al., 2017*; *Trapnell et al., 2014*). A Jupyter notebook that performs these analyses is in *Supplementary file 1*. For each sample, cell barcodes that had >2.5-fold fewer or more UMI counts mapping to cellular transcripts than the sample mean were excluded from downstream analyses (see red vertical lines in *Figure 3B*).

In order to determine an appropriate cutoff for how many reads a cell needed to contain in order to be classified as infected, we calculated the mean viral barcode purity across all cells that contained at least a given fraction of viral mRNA and had multiple viral reads that could be assigned a barcode (*Figure 4B* and *Figure 4—figure supplement 2*). We then determined the threshold fraction of viral mRNA at which the mean purity no longer increased as a function of the amount of viral

mRNA. This threshold represents the point at which we have effectively eliminated cells that have low barcode purity simply due to lysis-acquired reads sampled randomly from both viral barcodes. As is apparent from *Figure 4B*, only the 10 hr sample and the 8 hr-2 sample have the excess of mixed barcodes among cells with low amounts of viral mRNA. The likely reason is that these samples have more total viral mRNA (and so there is more available mRNA to be acquired from lysed cells); in addition, there is always some experimental variability in the amount of cell lysis during the 10X sequencing process, and these samples may simply have the most. So the above threshold procedure is appropriate for those two samples. For the other samples, we simply set a minimum threshold of requiring at least a fraction two× reads to come from viral mRNA as explained in the legend to *Figure 4—figure supplement 2*. The thresholds for each sample are shown in *Figure 4C* and *Figure 4—figure supplement 2*. This procedure is expected to be conservative, and may miss some truly infected cells with very low amounts of viral mRNA. For subsequent analyses, we retained all infected cells and a subsample of uninfected cells (the greater of 50 or the number of infected cells for that sample). The rationale for subsampling the uninfected cell is that the vast majority of cells are uninfected, and we did not want these cells to completely dominate the downstream analyses. Cells were classified as co-infected if both viral variants had an RNA level that exceeded the threshold, and if the minor variant contributed at least 5% of the viral mRNA.

For the semi-supervised t-SNE clustering, we used Monocle's cell hierarchy function to bin cells into those with no viral mRNA, <2% viral mRNA, between 2% and 20% viral mRNA, and >20%. Candidate marker genes for t-SNE dimensionality reduction were then determined using the Monocle function markerDiffTable, excluding the effects of sample variation and the number of genes identified in a given cell, using a q-value cutoff of 0.01. The specificity of these markers was determined using the function calculateMarkerSpecificity – the top 50 markers were retained, and used to place populations in a two-dimensional plane based on tSNE dimensionality reduction.

For the analyses of cellular genes that differed in expression as a function of the amount of viral mRNA, we only considered cells that expressed all eight viral mRNAs to avoid effects driven simply by viral gene absence. We also only considered cellular genes in the differential gene analysis, since viral gene expression will tautologically co-vary with the amount of viral mRNA. Additionally, because influenza virus has the capacity to degrade or prevent the synthesis of host mRNAs (*Bercovich-Kinori et al., 2016*) and contributes significantly to the total number UMIs in some cells, we calculate size factors (a scalar value representing efficiency of UMI capture) based on cellular transcripts alone. Finally, we assigned all cells a ceiling fraction of mRNA from virus of 25% so that a few extremely high-expressing cells did not dominate. Cellular genes with expression that co-varied with the fraction of viral mRNAs in a cell were then determined using the Monocle differentialGeneTest, after removing variance explained by sample to sample variation. *Figure 9* shows all genes that were significantly associated with the fraction of mRNA from virus at a false discovery rate of 0.1. We performed the gene set analysis using the P -alues from the Monocle differentialGeneTest with piano (*Väremo et al., 2013*) using the hallmark gene set from GSEA v6 (*Subramanian et al., 2005*) and Fisher's method.

## Acknowledgements

We thank Xiaojie Qiu for advice about use of the Monocle software package, David Bacsik and Robert Bradley for comments on the manuscript, and the Fred Hutch Genomics Core for performing the Illumina deep sequencing.

## Additional information

### Funding

| Funder | Grant reference number | Author |
| --- | --- | --- |
| National Institute of General Medical Sciences | R01GM102198 | Jesse D Bloom |
| National Institute of Allergy and Infectious Diseases | AI127897 | Jesse D Bloom |

| Damon Runyon Cancer Research Foundation | DRG-2227-15 | Alistair B Russell |
| Burroughs Wellcome Fund | Young Investigator in the Pathogenesis of Infectious Diseases | Jesse D Bloom |
| Simons Foundation | Faculty Scholar Award | Jesse D Bloom |
| Howard Hughes Medical Institute | Faculty Scholar Award | Jesse D Bloom |
| Eunice Kennedy Shriver National Institute of Child Health and Human Development | DP2OD020868 | Cole Trapnell |
| William Keck Foundation | Keck Foundation Grant | Cole Trapnell |
| Alfred P. Sloan Foundation | Sloan Research Fellowship | Cole Trapnell |

The funders had no role in study design, data collection and interpretation, or the decision to submit the work for publication.

## Author contributions
Alistair B Russell, Conceptualization, Formal analysis, Investigation, Writing—original draft, Writing—review and editing; Cole Trapnell, Conceptualization, Software, Formal analysis, Writing—review and editing; Jesse D Bloom, Conceptualization, Data curation, Software, Formal analysis, Funding acquisition, Writing—original draft, Writing—review and editing

## Author ORCIDs
Alistair B Russell [iD] http://orcid.org/0000-0002-5342-2309
Jesse D Bloom [iD] http://orcid.org/0000-0003-1267-3408

## Decision letter and Author response
Decision letter https://doi.org/10.7554/eLife.32303.044
Author response https://doi.org/10.7554/eLife.32303.045

# Additional files

## Supplementary files
• Supplementary file 1. Computer code for the analyses. This ZIP file contains a Jupyter notebook that runs CellRanger to align and annotate the reads, and a Jupyter notebook that uses Monocle to analyze the cell-gene matrix. The ZIP file also includes associated custom scripts. To just run the Monocle analysis in monocle_analysis.ipynb on a pre-generated cell-gene matrix, unpack *Supplementary file 2* into. /results/cellgenecounts/.
DOI: https://doi.org/10.7554/eLife.32303.036

• Supplementary file 2. The annotated cell-gene matrix in Matrix Market Format. This is the matrix generated in. /results/cellgenecounts/ by running the CellRanger analysis in align_and_annotate. ipynb in *Supplementary file 1*. This file is available on DataDryad at https://doi.org/10.5061/dryad. qp0t3.
DOI: https://doi.org/10.7554/eLife.32303.037

• Transparent reporting form
DOI: https://doi.org/10.7554/eLife.32303.038

## Data availability
The following datasets were generated:

| Author(s) | Year | Dataset title | Dataset URL | Database and Identifier |
|---|---|---|---|---|
| Russell AB, Trapnell C, Bloom JD | 2018 | Deep sequencing data | https://www.ncbi.nlm.nih.gov/geo/query/acc.cgi?acc=GSE108041 | Gene Expression Omnibus, GSE108041 |

Russell AB, Trapnell C, Bloom JD  2018  Annotated cell-gene matrix  http://dx.doi.org/10.5061/dryad.qp0t3  Dryad, 10.5061/dryad.qp0t3

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
