## [Decision Letter]

Thank you for submitting your article "Extreme heterogeneity of influenza virus infection in single cells" for consideration by *eLife*. Your article has been reviewed by two peer reviewers, and the evaluation has been overseen by a Reviewing Editor and Arup Chakraborty as the Senior Editor. The reviewers have opted to remain anonymous.

The reviewers have discussed the reviews with one another and the Reviewing Editor has drafted this decision to help you prepare a revised submission.

Summary:

Your manuscript describes the use of a novel single cell RNAseq method to examine heterogeneity in viral mRNA production within single cells infected with single influenza A virions. You report that viral mRNA levels, as a proportion of total cellular mRNA, can vary significantly between infected cells, and this variability cannot be simply explained by the activity of innate anti-viral defenses. Further, correlating the levels of specific host transcripts with viral transcript levels allows you to identify pathways that tend to correlate with viral levels.

The strengths of the paper include the description of a novel method for leveraging single cell cRNAseq to quantify heterogeneity and stochasticity in viral gene expression during infection. This is a really important question for the field, and the approaches detailed here will likely by adopted by other groups studying other viruses. The authors do a good job of accounting for many of the pitfalls inherent in the experimental execution and data analysis. The manuscript is well written, and does an especially effective job of visually presenting complex datasets.

The main shortcoming of the paper is that it provides very little in the way of new information. As the authors clearly and honestly point out, most of the findings in this paper are simply confirming observations made in older papers. Additionally, the findings are purely descriptive, and provide little insight into the mechanisms that may give rise to the observations.

We believe that addressing the points below would help ameliorate some of the shortcomings of the paper.

Major comments:

1) There are some obvious sources of variability that haven't been suitably discussed.

- The first is variation in the timing of the early stages of infection. No steps appear to have been taken to synchronize infection, or to limit secondary spread of the virus within the cultures. Could heterogeneity in the timing of binding/entry/fusion/trafficking explain a lot of the variation observed?

- Another potential contributor is the cell cycle status of the infected cells. Did the host transcript data shed light on whether cell cycle status influenced viral transcription levels? This is especially important to address given that some of the genes showing association with the viral burden are cell-cycle related.

- In assessing to what extent lack of RNP expression accounts for the viral mRNA expression variability (subsection “Absence of viral genes partially explains cell-to-cell variability in viral load”, second paragraph), we think it is important to take into account potential extracellular contamination. Specifically, it would be more convincing if the analysis omitted cells with mixed wildtype/synonymous clones.

- As the variability of the viral RNA load appears to be the central result of the manuscript, it would be useful to:a) Employ a statistic to quantify the variability (e.g. entropy or Gini index);b) Use simple models to illustrate how much variability can be accounted for by simple effects, such as expected Poisson co-infection frequency, or the likelihood of attaining full complement of RNP genes.

2) While the silent tagging method used to address co-infection is clever and appreciated, the issue is not fully settled. The dismissal of co-infection as a factor influencing cell-to-cell variation in the last paragraph of the subsection “Single cells show an extremely wide range of expression of viral mRNA”, is based on too few cells to draw any conclusions, and thus needs to be tempered. Also, there is likely to be a significant amount of cryptic co-infection with identical barcode viruses (expected to be similar to that of mixed barcodes, ~10%) that could influence measured heterogeneity. These points should be made in the text.

3) The analysis of host cell transcripts positively or negatively associated with high viral mRNA expression is pretty minimal. Do the host genes identified here match up with the results of previous studies that screened for host pro- and anti-viral factors (reviewed in Watanabe et al., 2010)? Also, targeted gene knockdown or over-expression experiments could help establish the causality underlying these relationships.

4) All analysis of viral gene expression seems to be at the segment level. How do different transcripts expressed from the same gene segment compare (i.e. NS1 and NEP)?

5) The determination of the minimum required influenza fraction (Figure 4) is based on sound logic; however, we are concerned that the observed results do not fully align with this model. Specifically, while the 10hr experiment looks reasonable, the distributions of wildtype/synonymous mixed cells in other experiments do not appear to show the same trend (most notable for 6hr and 8hr samples, where almost no mixed cells are observed at low fractions). In that regard, using the 10hr dataset to estimate thresholds for all of the other samples does not seem appropriate.

---

## [Author Response]

Summary:[…] The main shortcoming of the paper is that it provides very little in the way of new information. As the authors clearly and honestly point out, most of the findings in this paper are simply confirming observations made in older papers. Additionally, the findings are purely descriptive, and provide little insight into the mechanisms that may give rise to the observations.We believe that addressing the points below would help ameliorate some of the shortcomings of the paper.

This is a fair summary of our work. Thank you for recognizing the power of the approach and our efforts to fully describe the pitfalls and effectively present the complex datasets.

As you note, our approach does not uncover any truly new biological phenomenon, and (as you also note) we honestly describe how our results end up mostly confirming older observations made using more indirect methods. However, there is substantial value to explicitly observing these processes in single cells, even if the findings are mostly in line with existing circumstantial evidence from bulk studies or less comprehensive single-cell approaches (e.g., flow cytometry). Also, the extremeness of the heterogeneity across cells is surprising even in light of all that was previously known.

We have revised our manuscript in response to the many helpful suggestions from the reviewers. As you will see, we have addressed all the major questions and suggestions.

Major comments:1) There are some obvious sources of variability that haven't been suitably discussed.- The first is variation in the timing of the early stages of infection. No steps appear to have been taken to synchronize infection, or to limit secondary spread of the virus within the cultures. Could heterogeneity in the timing of binding/entry/fusion/trafficking explain a lot of the variation observed?

This is an important point. We did not synchronize infection, although for our 8-hour samples, we washed the cells one hour after infection for one sample (8hr) but not the other (8hr-2) in order to evaluate the importance of infection timing. The heterogeneity was similar between these two samples, as demonstrated by the new quantification method (Gini coefficients) suggested by the reviewers in a later comment.

We have also performed new experiments where we synchronized infection by pre-binding virus on ice, washing it away, and then raising the temperature. Analysis of this experiment by flow cytometry staining is in the new Figure 4—figure supplement 4, and shows that these synchronized infections look similar to ones that are neither synchronized nor washed.

Finally, the barcoded viruses (Figure 4B) provide strong evidence against secondary infection, since secondary infection would tend to lead to mixed-barcode cells because there would be many of progeny viruses of both barcodes.

In addition to adding Figure 4—figure supplement 4, we have added the following text related to this point:

“One possible source of heterogeneity in the amount of viral mRNA per cell is

variability in the timing of infection. […] Therefore, variability in the timing of infection is not the dominant cause of the cell-to-cell heterogeneity in our experiments.”

- Another potential contributor is the cell cycle status of the infected cells. Did the host transcript data shed light on whether cell cycle status influenced viral transcription levels? This is especially important to address given that some of the genes showing association with the viral burden are cell-cycle related.

We have expanded the analysis to specifically look at cell-cycle related genes. A geneset analysis (Figure 9—source data 2) finds that the G2-M cell-cycle checkpoint is associated with viral mRNA, with the expression of genes in this pathway typically expressed at lower levels in cells with more viral mRNA. This analysis is now discussed in the second-to-last paragraph of the Results.

- In assessing to what extent lack of RNP expression accounts for the viral mRNA expression variability (subsection “Absence of viral genes partially explains cell-to-cell variability in viral load”, second paragraph), we think it is important to take into account potential extracellular contamination. Specifically, it would be more convincing if the analysis omitted cells with mixed wildtype/synonymous clones.

This is a good suggestion. We have re-performed the analyses omitting the co-infected cells with mixed viral barcodes. These new analyses are in Figure 5—figure supplement 3 and Figure 5—figure supplement 5. The results remain essentially unchanged from the ones analyzing the full set of cells that are shown in main panels of Figure 5A, B.

- As the variability of the viral RNA load appears to be the central result of the manuscript, it would be useful to:a) Employ a statistic to quantify the variability (e.g. entropy or Gini index);b) Use simple models to illustrate how much variability can be accounted for by simple effects, such as expected Poisson co-infection frequency, or the likelihood of attaining full complement of RNP genes.

The idea of quantifying the variability in viral mRNA across infected cells is a good one. We have done this using the Gini coefficient. The results are in Figure 4—figure supplement 3, and are referred to in the text related to that figure. As a fun point of comparison, the Gini coefficients for the variability of viral mRNA across cells (≥0.64 for all samples) exceed the Gini coefficient for the unevenness of income distribution in the United States.

We have also simulated the expected distribution under a simple model where the heterogeneity is due only to Poisson co-infection and whether or not a cell has the full complement of RNP genes. This simulation and its results are in Figure 5—source data 2. The simulation gives a Gini coefficient as large as the actual one, but the shape of the simulated distribution is essentially bimodal, whereas the actual distribution is much more continuous and broad. Therefore, there are also other sources of heterogeneity that broaden the distribution beyond what is due only to RNP presence / absence and co-infection.

2) While the silent tagging method used to address co-infection is clever and appreciated, the issue is not fully settled. The dismissal of co-infection as a factor influencing cell-to-cell variation in the last paragraph of the subsection “Single cells show an extremely wide range of expression of viral mRNA”, is based on too few cells to draw any conclusions, and thus needs to be tempered. Also, there is likely to be a significant amount of cryptic co-infection with identical barcode viruses (expected to be similar to that of mixed barcodes, ~10%) that could influence measured heterogeneity. These points should be made in the text.

These are very good points. We have added the following text that clearly draws out the fact that our approach will only identify half the co-infections:

“In addition, the synonymous viral barcodes only identify co-infections by viruses with different barcodes – since the barcodes are at roughly equal proportion, we expect to miss about half of the co-infections. Since we annotate about ∼10% of the infected cells as co-infected by viruses with different barcodes (Figure 4D), we expect another ∼10% of the infected cells to also be co-infected but not annotated as so by our approach.”

The reviewers are correct that our statement about the role of co-infection in influencing cell-to-cell variation was too strong. We have clarified this statement. We have also added a figure supplement (Figure 4—figure supplement 5) which uses an independent method (flow cytometry staining of individual viral proteins) to support the fact that even at high infectious dose both low- and high-producing cells remain, although their relative proportions change. This figure supplement also supports our clarified statement that viral mRNA (or protein) production is not a simple continuous function of infectious dose, but is also affected by other sources of heterogeneity. The new text reads:

“Notably, Figure 4E shows that there are co-infected cells with both low and high amounts of viral mRNA, suggesting that the initial infectious dose does not drive a simple continuous increase in viral transcript production. […] This analysis shows that sub-populations of cells that express similarly low and high levels of viral proteins persist across a wide range of infectious doses, although co-infection can influence the relative proportion of infected cells that fall into these subpopulations (Figure 4—figure supplement 5).”

3) The analysis of host cell transcripts positively or negatively associated with high viral mRNA expression is pretty minimal. Do the host genes identified here match up with the results of previous studies that screened for host pro- and anti-viral factors (reviewed in Watanabe et al., 2010)? Also, targeted gene knockdown or over-expression experiments could help establish the causality underlying these relationships.

We have substantially expanded the last section of the Results that focuses on cellular gene expression. We call out the gene-set enrichments, which show that genes involved in oxidative stress responses are at higher levels in cells with more viral mRNA. The possible pro-viral role of transient oxidative stress has been reported before, and we have performed new experiments (Figure 9—figure supplement 2) that directly support the idea that it is pro-viral. These new experiments therefore suggest causality for one of the host-cell factors that we identify.

We also compare the genes that we find significantly associated with viral mRNA to the 128 genes highlighted by Watanabe et al. (2010) as having been found multiple times in genomewide studies. There is no overlap. This fact is discussed in the last paragraph of the Results. We theorize that this is because most of the cell-to-cell heterogeneity during infection of “normal” (e.g., not genetically perturbed) cells is *not* due to changes in expression of the type of large-effect genes that are typically identified in genome-wide screens. Rather, our results would seem to suggest that a substantial amount of the heterogeneity is due to viral / stochastic factors (e.g., failure to express certain viral genes), or perhaps more subtle changes in cellular state such as transient oxidative stress.

Importantly, we do *not* believe that the main value of our approach is as a method to identify pro- or anti-viral genes. Our goal is to figure out the actual extent and sources of cell-to-cell heterogeneity, which may not be due to primarily alterations in the expression level of single host genes with large effects.

4) All analysis of viral gene expression seems to be at the segment level. How do different transcripts expressed from the same gene segment compare (i.e. NS1 and NEP)?

This is an important point that we failed to adequately discuss in the original manuscript. As the reviews note, two of the influenza transcripts (M1 / M2 and NS1 / NS2) are generated from alternative splicing of gene segments. However, each pair of these transcripts share the same 3’ end. Most current single-cell mRNA sequencing strategies, including the 10X platform that we use, can only sequence the 3’ end of the transcript. The reason is that the cell barcode is appended to this end, and so Illumina sequencing can only extend to regions proximal to the 3’ end if they are still to capture the barcode (as is required for single-cell mRNA sequencing). Therefore, these techniques cannot accurately identify alternative splicing – this is simply an inherent limitation of the current techniques. We now clearly explain this fact in the following added text:

“Note that influenza virus expresses ten major gene transcripts from its eight gene segments, as the M and NS segments are alternatively spliced to produce the M1 / M2 and NS1 / NEP transcript, respectively (Dubois et al., 2014). However, an inherent limitation of current established single-cell mRNA sequencing techniques is that they only sequence the 3’ end of the transcript (Zheng et al., 2017; Macosko et al., 2015; Klein et al., 2015; Cao et al., 2017). Since the alternative spliceoforms M1 / M2 and NS1 / NEP share the same 3’ ends, we cannot distinguish them and therefore will refer simply to the combined counts of transcripts from each of these alternatively spliced segments as the M and NS genes.”

5) The determination of the minimum required influenza fraction (Figure 4) is based on sound logic; however, we are concerned that the observed results do not fully align with this model. Specifically, while the 10hr experiment looks reasonable, the distributions of wildtype/synonymous mixed cells in other experiments do not appear to show the same trend (most notable for 6hr and 8hr samples, where almost no mixed cells are observed at low fractions). In that regard, using the 10hr dataset to estimate thresholds for all of the other samples does not seem appropriate.

This is a good point. We now use the procedure in Figure 4C to call thresholds *separately* for each sample as the reviewers suggest. The sample-specific calling is shown in Figure 4—figure supplement 3. These new thresholds lead to a modest increase in the number of cells called as virally infected for the earlier timepoint samples. These newly called cells therefore slightly shift the quantitative values in all subsequent analyses, although the changes are very small and all results remain unchanged.

The reason that the earlier timepoints have less mixed barcodes is because these earlier samples have much less total viral mRNA – and when there is less viral mRNA in total, there is also less available to be acquired from lysed cells.